# Seeing Through Words: Controlling Visual Retrieval Quality with Language Models

**Jianglin Lu**[1,2*]  **Simon Jenni**[1]  **Kushal Kafle**[1]  **Jing Shi**[1]  **Handong Zhao**[1]  **Yun Fu**[2]
[1]Adobe Research   [2]Northeastern University

## Abstract

Text-to-image retrieval is a fundamental task in vision-language learning, yet in real-world scenarios it is often challenged by short and underspecified user queries. Such queries are typically only one or two words long, rendering them semantically ambiguous, prone to collisions across diverse visual interpretations, and lacking explicit control over the quality of retrieved images. To address these issues, we propose a new paradigm of *quality-controllable retrieval*, which enriches short queries with contextual details while incorporating explicit notions of image quality. Our key idea is to leverage a generative language model as a query completion function, extending underspecified queries into descriptive forms that capture fine-grained visual attributes such as pose, scene, and aesthetics. We introduce a general framework that conditions query completion on discretized quality levels, derived from relevance and aesthetic scoring models, so that query enrichment is not only semantically meaningful but also quality-aware. The resulting system provides three key advantages: ① *flexibility*, it is compatible with any pretrained vision-language model (VLMs) without modification; ② *transparency*, enriched queries are explicitly interpretable by users; and ③ *controllability*, enabling retrieval results to be steered toward user-preferred quality levels. Extensive experiments demonstrate that our proposed approach significantly improves retrieval results and provides effective quality control, *bridging the gap between the expressive capacity of modern VLMs and the underspecified nature of short user queries*. Our code is available at https://github.com/Jianglin954/QCQC.

## 1 Introduction

Text-to-image retrieval (T2IR) aims to return the most relevant images from a gallery given a textual query. Recent progress in this task has been largely driven by vision–language models (VLMs) (Jia et al., 2021; Yang et al., 2022; Li et al., 2023; Yang et al., 2025; Lu et al., 2025a; Huang et al., 2025), which learn joint representations of text and images through large-scale pretraining on web-scale image–text pairs (Schuhmann et al., 2021; 2022; Liu et al., 2023). These models significantly narrow the semantic gap between modalities and achieve strong alignment across diverse benchmarks (Ilharco et al., 2021; Singh et al., 2022; Li et al., 2024; Lu et al., 2025b; Dong et al., 2026).

Despite these advances, retrieval performance often degrades in realistic scenarios where user queries are very short (typically just one or two words, e.g., "a dog"). Short queries encode only limited semantics, which results in large and ambiguous search subspaces and less discriminative results. This issue becomes more pronounced in large-scale galleries, where underspecified queries yield many candidate matches and cause semantic collisions among visually diverse results.

Another limitation of existing retrieval systems is their singular focus on semantic alignment. Naïve retrieval approaches simply return the top-$k$ images with the highest similarity scores, overlooking other critical aspects of user satisfaction such as aesthetics, interestingness, or popularity (Yi et al., 2023; Abdullahu & Grabner, 2024; Wang et al., 2025). In practice, *retrieval quality* is context-dependent: art students may prefer visually inspiring images, architects may seek unique and creative references, and shoppers may favor popular or visually appealing products. However, conventional retrieval systems lack mechanisms for steering retrieval toward these quality dimensions.

---

*Corresponding author: jianglinlu@outlook.com. Work done during an internship at Adobe Research.

To address these limitations, we introduce the task of *quality-controllable retrieval* (QCR). Formally, given a frozen VLM and a short textual query, the objective is to retrieve images that not only align semantically but also satisfy user-specified quality requirements. This setting is feasible because short queries occupy a broad region of the embedding space that contains images of varying perceptual quality. With appropriate conditioning, this region can be partitioned into perceptually distinct subsets, enabling fine-grained quality-aware retrieval.

In this work, we define retrieval quality along two widely applicable dimensions: *relevance* (semantic consistency) (Cherti et al., 2023) and *aesthetics* (visual appeal) (Yi et al., 2023). For each image in the gallery, we construct auxiliary annotations consisting of a textual description, a relevance score, and an aesthetic score. We discretize these continuous scores into categorical quality levels and associate each description with its corresponding quality condition.

The central challenge is how to steer retrieval results toward specific quality levels given short queries. We propose a simple yet effective solution: *quality-conditioned query completion* (QCQC). QCQC enriches short queries with quality-aware details by leveraging a generative large language model (LLM). Trained on the quality-augmented dataset, the LLM learns to append appropriate descriptive phrases that capture both semantic and quality-related attributes. By conditioning on distinct quality levels, QCQC generates targeted query completions that steer retrieval toward specific regions of the embedding space. This capability is particularly valuable in practice, as users often struggle to articulate quality preferences or may lack a clear understanding of what constitutes "high" or "low" quality within a dataset. By modeling how textual descriptions vary across quality levels, our approach bridges this gap and enables more controllable, quality-aware retrieval through conditioned query completion. Our key contributions in this work can be summarized as follows:

- A new problem: we introduce quality-controllable retrieval, a new setting where retrieval can be explicitly conditioned on user-defined quality requirements.
- A general solution: we propose QCQC, a generative query completion framework that leverages LLMs to enrich short queries with quality-aware descriptive details.
- Validation: we conduct extensive experiments to show that QCQC effectively steers retrieval outcomes according to quality preferences and is compatible to multiple VLMs.

## 2 PRELIMINARIES

### 2.1 MOTIVATION

We study the problem of text-to-image retrieval, where the goal is to return the desired images from a large gallery given a set of natural language queries. Specifically, let $\mathcal{Q} := \{Q_1, \ldots, Q_m\}$ denote a collection of $m$ text queries and $\mathcal{I} := \{I_1, \ldots, I_n\}$ an image gallery of size $n$. We consider a state-of-the-art VLM as the retrieval backbone, equipped with a text encoder $g : \mathcal{Q} \to \mathbb{R}^d$ and an image encoder $f : \mathcal{I} \to \mathbb{R}^d$, both producing $d$-dimensional normalized embeddings. Given a query set $\mathcal{Q}$, the system returns the top-$\eta$ relevant images according to

$$\mathcal{X} := \text{sort}\left(f(\mathcal{I}),\ g(\mathcal{Q}),\ \eta\right), \tag{1}$$

where $\mathcal{X} \subseteq \mathcal{I}$ denotes the top-$\eta$ matches of queries $\mathcal{Q}$. The sort function typically operates on the similarity scores $\boldsymbol{S} \in \mathbb{R}^{m \times n}$ with $\boldsymbol{S}_{ij} := g(Q_i)^\top f(I_j)$.

Although modern VLMs achieve strong cross-modal alignment, retrieval performance deteriorates in realistic scenarios where user queries are usually very short (typically just one or two words, e.g., "a dog"). Given such short queries, naïve retrieval system faces several challenges: ① *Semantic ambiguity*: a few words can refer to a wide range of possible images, leading to a large and diffuse search subspace with less discriminative retrieval results. ② *Semantic collisions*: short queries tend to yield close similarity scores for visually diverse images. These collisions confuse ranking and are particularly problematic in large-scale galleries where many candidate images match the vague query. ③ *Lack of quality control*: the quality of retrieved images is not explicitly enforced during retrieval. At best, one can apply post-retrieval filtering, but the system itself provides no mechanism to ensure that high-quality results consistently appear among the top matches. These issues highlight *a fundamental gap between the expressive capacity of modern VLMs and the underspecified nature of user queries*, motivating the need for query enrichment and controllable retrieval mechanisms.

## 2.2 PROBLEM SETTING

To address the above limitations, we propose to enrich short queries with additional descriptive details that potentially capture more distinguishable attributes of images. Formally, let $h$ denote a query completion function that maps $\mathcal{Q}$ to enriched queries $h(\mathcal{Q})$. Retrieval is then performed as

$$\widetilde{\mathcal{X}} := \text{sort}\left(f(\mathcal{I}),\ g(h(\mathcal{Q})),\ \eta\right), \tag{2}$$

where $h(\mathcal{Q})$ augments the short queries with contextual details. The enriched queries are expected to capture not only object categories but also additional information such as pose, scene, action, and fine-grained attributes. To be effective, the completion function *should be aware of the retrieval gallery, so that it generates meaningful context rather than irrelevant content*.

To achieve this, we implement $h$ using a generative large language model (LLM). However, simply training the LLM on image descriptions is insufficient, since it cannot guarantee that retrieval results satisfy user expectations of quality. Instead, we partition the textual descriptions into non-overlapped quality levels $\mathcal{C}$ that reflect different image quality categories. We then finetune the LLM with these quality levels, enabling it to generate query completions conditioned on quality preferences. This yields the formulation of our *quality-controllable retrieval* (QCR) :

$$\widetilde{\mathcal{X}} := \text{sort}\left(f(\mathcal{I}),\ g(\texttt{LLM}(\mathcal{Q}\,;\,\mathcal{C})),\ \eta\right), \tag{3}$$

where $\texttt{LLM}(\mathcal{Q};\mathcal{C})$ expands the short queries based on the specified quality constraint $\mathcal{C}$. The extended queries thus steer retrieval toward images that align with the desired quality criteria.

This approach offers several practical benefits: ① *Flexibility*: it requires no modification to pre-trained VLMs and remains compatible with any VLMs; ② *Transparency*: the generated query completions are human-readable, allowing users to review and select preferred options. ③ *Controllability*: the LLM can produce different query completions according to distinct quality conditions $\mathcal{C}$, enabling explicit quality control during retrieval. In the following section, we provide theoretical justification for why enriching short queries may improve retrieval performance.

## 2.3 THEORETICAL ANALYSIS

We model query completion as a structured perturbation and analyze its effect on the similarity matrices $\boldsymbol{S}$ through the lens of rank variation under perturbations.

Let $\mathcal{Q}^+ = \{Q_1^+, \ldots, Q_m^+\} := h(Q)$ denote the extended queries by $h$, where $Q_i^+ := Q_i + \text{suffix}_i$, $\forall i \in \{1, \ldots, m\}$, and $\text{suffix}_i$ denotes additional descriptive details appended to query $Q_i$. Let $\boldsymbol{C} \in \mathbb{R}^{n \times d}$ be the image embedding matrix with rows $\boldsymbol{c}_j := f(I_j) \in \mathbb{R}^d$, $\forall j \in \{1, \ldots, n\}$, and $\boldsymbol{A}, \boldsymbol{B} \in \mathbb{R}^{m \times d}$ be two sets of text embeddings with a strict one-to-one pairing of rows, with rows $\boldsymbol{a}_i := g(Q_i) \in \mathbb{R}^d$ and $\boldsymbol{b}_i := g(Q_i^+) \in \mathbb{R}^d$, $\forall i \in \{1, \ldots, m\}$. Let $r := \text{rank}(\boldsymbol{A})$ be the rank of $\boldsymbol{A}$, $\sigma_r(\boldsymbol{A})$ be the smallest nonzero singular value of $\boldsymbol{A}$, and $\boldsymbol{A} = \boldsymbol{U}\boldsymbol{\Sigma}\boldsymbol{V}^\top$ denote its singular value decomposition (SVD). We then partition the right singular vectors as $\boldsymbol{V} = \begin{bmatrix} \boldsymbol{V}_S & \boldsymbol{V}_\perp \end{bmatrix}$, where $\boldsymbol{V}_S \in \mathbb{R}^{d \times r}$ and $\boldsymbol{V}_\perp \in \mathbb{R}^{d \times (d-r)}$ satisfy $\text{span}(\boldsymbol{V}_S) = \mathcal{R}(\boldsymbol{A})$ and $\text{span}(\boldsymbol{V}_\perp) = \mathcal{R}(\boldsymbol{A})^\perp$, with $\mathcal{R}(\boldsymbol{A}) := \text{span}\{\boldsymbol{a}_1^\top, \ldots, \boldsymbol{a}_m^\top\} \subseteq \mathbb{R}^d$ the row space of $\boldsymbol{A}$.

**Definition 1.** *We define a perturbation matrix* $\boldsymbol{\Delta} := \boldsymbol{B} - \boldsymbol{A} \in \mathbb{R}^{m \times d}$, *score matrices* $\boldsymbol{S}_A := \boldsymbol{A}\boldsymbol{C}^\top \in \mathbb{R}^{m \times n}$ *and* $\boldsymbol{S}_B := \boldsymbol{B}\boldsymbol{C}^\top \in \mathbb{R}^{m \times n}$ *for the queries* $\mathcal{Q}$ *and* $\mathcal{Q}^+$, $\boldsymbol{A}_S := \boldsymbol{A}\boldsymbol{V}_S$, $\boldsymbol{\Delta}_S := \boldsymbol{\Delta}\boldsymbol{V}_S$, $\boldsymbol{\Delta}_\perp := \boldsymbol{\Delta}\boldsymbol{V}_\perp$, $\boldsymbol{C}_S := \boldsymbol{C}\boldsymbol{V}_S$, $\boldsymbol{C}_\perp := \boldsymbol{C}\boldsymbol{V}_\perp$, $\boldsymbol{X} := (\boldsymbol{A}_S + \boldsymbol{\Delta}_S)\boldsymbol{C}_S^\top$, $\boldsymbol{Y} := \boldsymbol{\Delta}_\perp \boldsymbol{C}_\perp^\top$, $\mathcal{U} := \text{col}(\boldsymbol{X})$, *and* $\boldsymbol{P} := \boldsymbol{P}_X$ *as the orthogonal projector onto* $\mathcal{U}$.

**Lemma 1.** *If* $\text{rank}(\boldsymbol{X}_I) = r$ *and* $\|\boldsymbol{X}_I^\dagger \boldsymbol{P} \boldsymbol{Y}_I\|_2 < 1$, *then* $\text{rank}(\boldsymbol{X}_I + \boldsymbol{P}\boldsymbol{Y}_I) = r$.

**Proposition 1.** *Assume that: i)* $\|\boldsymbol{\Delta}\|_2 < \sigma_r(\boldsymbol{A})$; *ii) there exists* $I \subseteq \{1, \ldots, n\}$ *with* $|I| = r$ *such that the columns* $\boldsymbol{X}_I$ *form a basis of* $\mathcal{U}$; *iii)* $\|\boldsymbol{X}_I^\dagger \boldsymbol{P} \boldsymbol{Y}_I\|_2 < 1$; *and iv) there exists disjoint index set* $K \subseteq \{1, \ldots, n\} \setminus I$ *such that* $k := \text{rank}\left((\boldsymbol{I} - \boldsymbol{P}_{\boldsymbol{Z}_I})\boldsymbol{Z}_K\right) \geq 1$, *where* $\boldsymbol{Z} := (\boldsymbol{I} - \boldsymbol{P})\boldsymbol{Y}$, $\boldsymbol{Z}_I := \boldsymbol{Z}_{:,I}$, $\boldsymbol{Z}_K := \boldsymbol{Z}_{:,K}$, *and* $\boldsymbol{P}_{\boldsymbol{Z}_I}$ *be the orthogonal projector onto* $\text{col}(\boldsymbol{Z}_I)$. *Then,* $\text{rank}(\boldsymbol{S}_B) > \text{rank}(\boldsymbol{S}_A)$.

This proposition indicates that, under certain conditions, the score matrix $\boldsymbol{S}_B$ derived from query completion can express more independent scoring patterns and has the ability to potentially make finer-grained distinctions. The detailed proof and analysis are provided in the Appendix A.1.

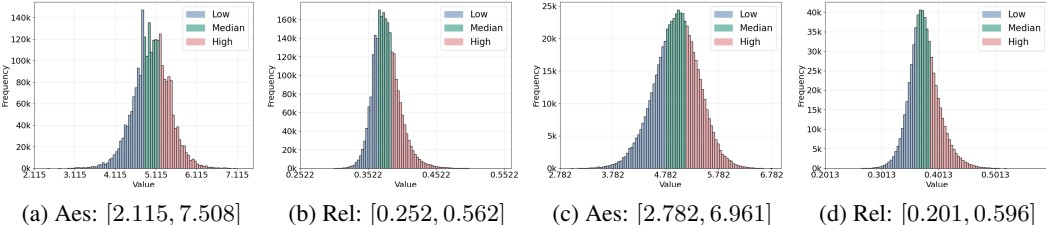

(a) Aes: [2.115, 7.508]  (b) Rel: [0.252, 0.562]  (c) Aes: [2.782, 6.961]  (d) Rel: [0.201, 0.596]

Figure 1: Aesthetic and relevance score distributions of `Flickr2.4M` in (a) and (b), and of `MS-COCO` in (c) and (d). It is worth noting that the numbers of high-quality and low-quality images are limited, which leads to the average scores of any two random sets being very close.

## 3  QUALITY-CONDITIONED QUERY COMPLETION

### 3.1  QUALITY DEFINITION

For the proposed QCR task, we require a clear notion of *quality*. In this work, we characterize quality along two primary dimensions: ① *Relevance*, which measures the semantic consistency between textual queries and their corresponding images; and ② *Aesthetics*, which reflects the visual appeal or attractiveness of retrieved images. Note that the proposed QCQC framework is inherently flexible, permitting the incorporation of arbitrary quality metric, provided that reliable scoring models are available and applicable to general image datasets. Other notions of quality, such as *interestingness* (Gygli et al., 2013; Abdullahu & Grabner, 2024) can also be adopted in a similar manner and are left for future exploration. To facilitate user control over retrieved results, we discretize the quality dimensions into non-overlapping conditions. Specifically, we define $\mathcal{C}^R$ for relevance and $\mathcal{C}^A$ for aesthetics, each partitioned into perceptually distinct and user-friendly levels. For example, both can be represented as $\mathcal{C}^R, \mathcal{C}^A := \{\texttt{Low}, \texttt{Medium}, \texttt{High}\}$.

### 3.2  DATA GENERATION

To ensure that the completion function can perceive the retrieval gallery, we construct an augmented training dataset for each gallery $\mathcal{I}$. The dataset integrates three complementary components: textual descriptions $\mathcal{T} = \{T_i\}_{i=1}^n$ of images, relevance scores $\boldsymbol{s}^r \in \mathbb{R}^n$, and aesthetic scores $\boldsymbol{s}^a \in \mathbb{R}^n$.

**Textual Descriptions.** For each image $I_i$, we generate a textual description $T_i$ using an image caption model $\texttt{CAP}(\cdot)$, i.e., $T_i = \texttt{CAP}(I_i), \forall i \in \{1, \ldots, n\}$. In our experiments, we utilize strong pretrained captioning models without additional fine-tuning for description generation. Each $T_i$ is a concise sentence summarizing the main content of the image.

**Aesthetic Scores.** We assign an aesthetic score $\boldsymbol{s}_i^a$ to each image $I_i$ using an aesthetic evaluation model $\texttt{EV}_A(\cdot)$, i.e., $\boldsymbol{s}_i^a = \texttt{EV}_A(I_i), \forall i \in \{1, \ldots, n\}$. The aesthetic scores represent the visual quality of the images, with higher scores indicating greater visual appeal.

**Relevance Scores.** For each image-description pair $\{I_i, T_i\}$, we compute a relevance score using a pretrained VLM. Specifically, we extract the image feature $f(I_i)$ and text feature $g(T_i)$, then calculate their cosine similarity as their relevance score, i.e., $\boldsymbol{s}_i^r = \texttt{cos}(f(I_i), g(T_i)), \forall i \in \{1, \ldots, n\}$.

### 3.3  TRAINING FRAMEWORK

**Score Discretization.** To simulate quality-controlled retrieval, we discretize the continuous quality scores of images into categorical levels that are more intuitive for users. Given a score vector $\boldsymbol{r}$ (either aesthetics $\boldsymbol{s}^a$ or relevance $\boldsymbol{s}^r$), each score $\boldsymbol{r}_i$ is mapped into one of three descriptive levels by partitioning the score distribution into three percentiles:[1]

$$l(\boldsymbol{r}_i) = \begin{cases} \texttt{Low}, & \boldsymbol{r}_i \leq \texttt{perc}(\boldsymbol{r}, p_1), \\ \texttt{High}, & \boldsymbol{r}_i > \texttt{perc}(\boldsymbol{r}, p_2), \\ \texttt{Medium}, & \text{otherwise}. \end{cases} \tag{4}$$

---

[1] Our framework is general and supports arbitrary numbers of levels depending on the desired granularity. In Table 5, Sec. 4.5, we provide an example with five quality levels.

Table 1: Query completions with their retrieved images and quality scores on MS-COCO

| Rel: Low, Aes: Low | Rel: Medium, Aes: Medium | Rel: High, Aes: High |
|---|---|---|
| 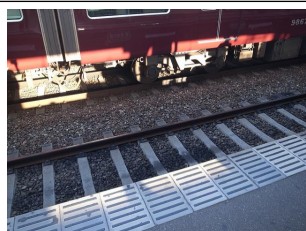 | 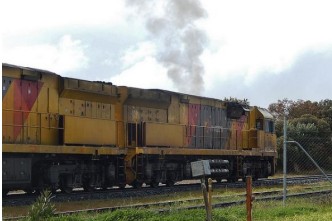 | 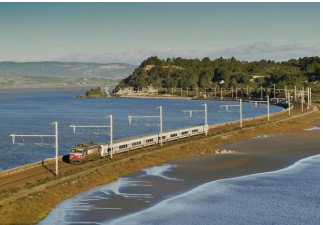 |
| **a train** *that is sitting on the tracks in gravel* 
 Aes 4.715, Rel 0.347 | **a train** *sitting on the tracks with black smoke coming out of it* 
 Aes 4.818, Rel 0.382 | **a train** *is traveling near some water and houses* 
 Aes 5.935, Rel 0.394 |
| 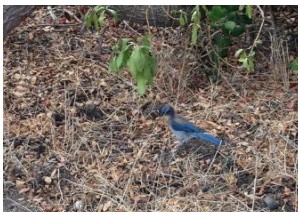 | 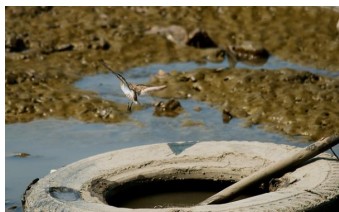 | 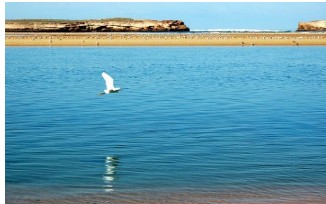 |
| **a bird** *standing on the ground near some leaves* 
 Aes 4.616, Rel 0.346 | **a bird** *flying above some brown water* 
 Aes 5.079, Rel 0.374 | **a bird** *flying across some water at the beach* 
 Aes 5.120, Rel 0.386 |
| 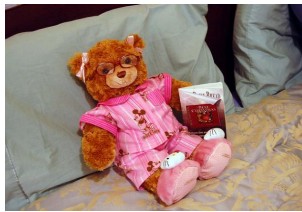 | 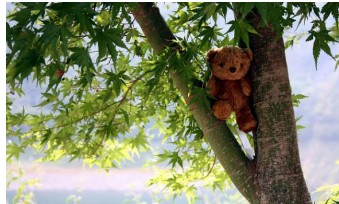 | 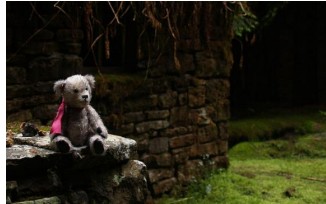 |
| **a teddy bear** *wearing eye glasses and laying on a bed* 
 Aes 4.788, Rel 0.359 | **a teddy bear** *that is sitting on a tree* 
 Aes 5.649, Rel 0.388 | **a teddy bear** *sitting on a wall next to an old stone house* 
 Aes 5.818, Rel 0.437 |

Here, $\boldsymbol{r}_i$ is the score of the $i$-th sample and $\texttt{perc}(\boldsymbol{r}, p)$ calculates the $p\%$ percentile of $\boldsymbol{r}$ as

$$\texttt{perc}(\boldsymbol{r}, p) = \widetilde{\boldsymbol{r}}[\lfloor \xi \rfloor] + (\xi - \lfloor \xi \rfloor) \cdot \left( \widetilde{\boldsymbol{r}}[\lfloor \xi \rfloor + 1] - \widetilde{\boldsymbol{r}}[\lfloor \xi \rfloor] \right), \tag{5}$$

where $\xi = \frac{p}{100} \cdot (n-1)$, $\widetilde{\boldsymbol{r}}$ is the sorted version of $\boldsymbol{r}$, and $\lfloor \cdot \rfloor$ is the floor function. Figure 1 illustrates the distributions of aesthetics and relevance scores and their discretized partitions.

**Instruction Design.** We then train the completion function LLM on the augmented training set $\mathcal{D} = \{\mathcal{T}, \boldsymbol{s}^a, \boldsymbol{s}^r\}$. The discretized quality levels serve as explicit conditions within instructions, enabling the LLM to generate quality-aware query completions. For each image $I_i$, we design a concise instruction $P_i$ of the following form:

$$\texttt{"Relevance: } l(\boldsymbol{s}_i^r)\texttt{, Aesthetic: } l(\boldsymbol{s}_i^a)\texttt{, Query: "}$$

where $l(\boldsymbol{s}_i^r)$ and $l(\boldsymbol{s}_i^a)$ represent the categorical quality levels defined in Eq. (4). During training, this instruction provides a lightweight yet effective mechanism to condition query generation on the specified quality preferences.

**Model Training.** To stimulate the quality control process during model training, we use the descriptive levels $l(\boldsymbol{s}_i^r)$ and $l(\boldsymbol{s}_i^a)$ of image $I_i$ as the quality conditions, which are incorporated into the instruction $P_i$. We then concatenate instructions $P_i$ with the textual description $T_i$ for each image $I_i$, and then train the proposed QCQC model with the standard autoregressive next-token predic-

Table 2: Query completions with their retrieved images and quality scores on `Flickr2.4M`

| Aes: **Low**, Rel: **Low** | Aes: **Medium**, Rel: **Medium** | Aes: **High**, Rel: **High** |
|---|---|---|
| 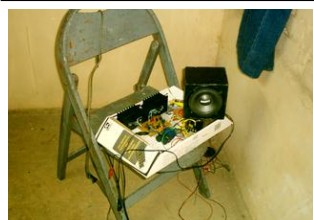 | 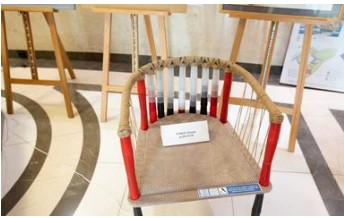 | 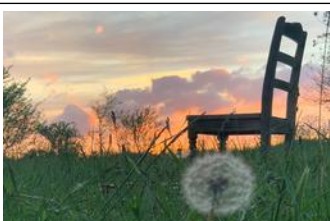 |
| ***a chair*** *with wires on it* 
 Aes 4.019, Rel 0.362 | ***a chair*** *with red and black ropes on it* 
 Aes 4.847, Rel 0.379 | ***a chair*** *on a stage in a field* 
 Aes 5.257, Rel 0.387 |
| 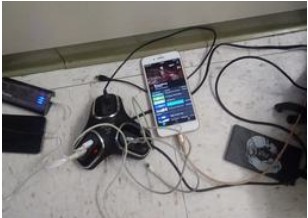 | 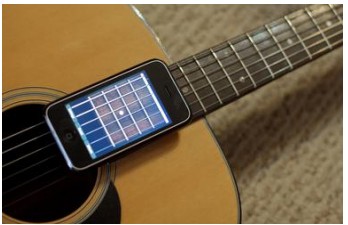 | 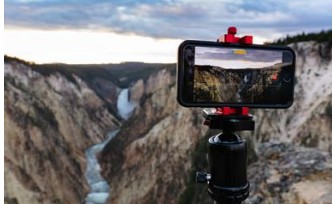 |
| ***a cell phone*** *with wires attached to it* 
 Aes 4.531, Rel 0.362 | ***a cell phone*** *with an acoustic guitar on it* 
 Aes 5.035, Rel 0.390 | ***a cell phone*** *on a tripod in front of a waterfall in yellowstone national park* 
 Aes 5.441, Rel 0.429 |
| 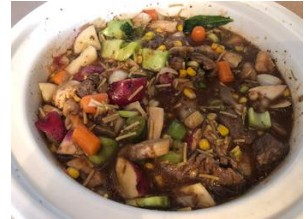 | 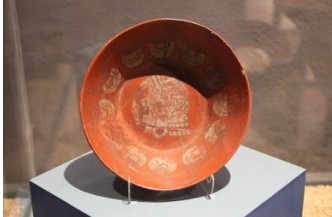 | 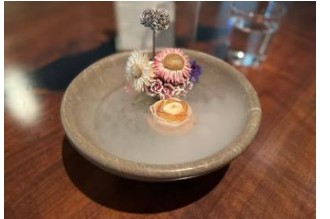 |
| ***a bowl*** *of soup with meat and vegetables in it* 
 Aes 4.648, Rel 0.343 | ***a bowl*** *on display* 
 Aes 4.980, Rel 0.379 | ***a bowl*** *with flowers on it* 
 Aes 5.386, Rel 0.387 |

tion loss. In this way, our QCQC model learns to generate query completions that are not only semantically relevant but also controllable according to the given quality constraints.

**Inference Strategy.** During the inference stage, we concatenate a similar instruction with each testing query. To simulate user preferences, we evaluate various relevance-aesthetic combinations, such as "low relevance, low aesthetic" and "high relevance, high aesthetic". Then the model generates completed queries based on the instructions, testing queries, and specified quality conditions. For efficient similarity search on large-scale galleries, we utilize the `FAISS` library (Johnson et al., 2019) to identify the nearest images for the queries.

## 4 EXPERIMENTS

### 4.1 EXPERIMENTAL SETTINGS

**Datasets.** We evaluate our method on two image datasets: one with real textual descriptions and one without. For the image-only one, we construct a large dataset sourced from the Openverse website (Openverse, 2025). We refer to this dataset as `Flickr2.4M`, which contains over 2.4 million CC0-licensed images randomly selected from the Flickr subset of Openverse. For image

Table 3: Retrieval quality of various methods on `Flickr2.4M`. CoCa and Blip2 are used to generate textual descriptions; **L** (Low), **M** (Medium), and **H** (High) indicate the quality conditions; and Ctrl specifies whether the method enables controllable retrieval over quality. For both average relevance (Ave Rel) and average aesthetics (Ave Aes), higher values indicate better retrieval quality.

| Quality | VLM | Aes Cond / Rel Cond | L / L | L / M | L / H | M / L | M / M | M / H | H / L | H / M | H / H | Ctrl ? |
|---|---|---|---|---|---|---|---|---|---|---|---|---|
| Prefix | −− | Ave Aes | 4.735 | 4.735 | 4.735 | 4.735 | 4.735 | 4.735 | 4.735 | 4.735 | 4.735 | × |
| | | Ave Rel | 0.350 | 0.350 | 0.350 | 0.350 | 0.350 | 0.350 | 0.350 | 0.350 | 0.350 | |
| LLaMA3 | −− | Ave Aes | 4.730 | 4.822 | 4.831 | 4.823 | 4.837 | 4.784 | 4.798 | 4.722 | 4.842 | × |
| | | Ave Rel | 0.351 | 0.351 | 0.351 | 0.353 | 0.351 | 0.350 | 0.354 | 0.354 | 0.352 | |
| GPT-4o | −− | Ave Aes | 4.359 | 4.651 | 4.728 | 4.712 | 4.668 | 4.791 | 4.791 | 4.816 | 5.056 | × |
| | | Ave Rel | 0.378 | 0.361 | 0.357 | 0.358 | 0.360 | 0.356 | 0.361 | 0.357 | 0.361 | |
| PT | −− | Ave Aes | 4.776 | 4.556 | 4.722 | 4.811 | 4.781 | 4.757 | 4.693 | 4.751 | 4.746 | × |
| | | Ave Rel | 0.345 | 0.346 | 0.349 | 0.349 | 0.346 | 0.348 | 0.345 | 0.350 | 0.350 | |
| FT | CoCa | Ave Aes | 4.756 | 4.834 | 4.777 | 4.838 | 4.863 | 4.882 | 4.821 | 4.905 | 4.770 | × |
| | | Ave Rel | 0.365 | 0.368 | 0.364 | 0.363 | 0.369 | 0.368 | 0.365 | 0.364 | 0.365 | |
| Ours | CoCa | Ave Aes | 4.458 | 4.615 | 4.530 | 4.934 | 4.852 | 4.841 | 5.222 | 5.170 | 5.270 | √ |
| | | Ave Rel | 0.355 | 0.366 | 0.391 | 0.354 | 0.360 | 0.386 | 0.353 | 0.368 | 0.390 | |
| FT | Blip2 | Ave Aes | 4.795 | 4.871 | 4.890 | 4.894 | 4.844 | 4.856 | 4.901 | 4.847 | 4.888 | × |
| | | Ave Rel | 0.370 | 0.368 | 0.367 | 0.367 | 0.371 | 0.366 | 0.367 | 0.371 | 0.369 | |
| Ours | Blip2 | Ave Aes | 4.541 | 4.523 | 4.455 | 4.940 | 4.906 | 4.922 | 5.309 | 5.222 | 5.191 | √ |
| | | Ave Rel | 0.353 | 0.370 | 0.397 | 0.354 | 0.366 | 0.396 | 0.355 | 0.372 | 0.390 | |

datasets with real textual descriptions, we adopt the widely-used `MS-COCO` dataset for experiments, which includes both images and human-annotated descriptions. Specifically, we utilize the training subset of `MS-COCO`, which consists of $118,287$ samples, each sample containing one image and five corresponding descriptions. In total, approximately $0.6$ million descriptions are used for training.

**Model Selection.** For the backbone of our method, we evaluate two different LLMs: GPT2-1.5B (Radford et al., 2019) and Qwen2.5-0.5B (Yang et al., 2025). Other LLMs can be validated similarly and we leave them for future study. We adopt two caption models $CAP(\cdot)$, including a pretrained CoCa (Yu et al., 2022) and a pretrained Blip2 (Li et al., 2023) model. For feature extraction, we adopt a pretrained VLM OpenCLIP (ViT-H-14-quickgelu) (Cherti et al., 2023; Ilharco et al., 2021). The relevance score is computed as the cosine similarity between the features of each image-description pair. For $EV_A(\cdot)$, we use a pretrained aesthetic predictor (Schuhmann, 2022).

**Implementation Details.** For GPT2-1.5B (Radford et al., 2019), we set the learning rate, warmup steps, number of epochs, and batch size to $2e-3$, $100$, $50$, $150$, respectively. For Qwen2.5-0.5B (Yang et al., 2025), these hyperparameters are set to $2e-5$, $100$, $30$, and $80$, respectively. For score discretization, we set $p_1 = 33$ and $p_2 = 66$ to divide the score distribution into three evenly spaced percentiles (examples of five-level cases are also considered). Note that we only train the query completion model LLM, while the quality evaluation model $EV_A(\cdot)$, the caption models $CAP(\cdot)$, and the retrieval model VLM are all pretrained without additional fine-tuning. Since the pretrained caption models may occasionally generate non-English characters, we clean these characters directly before training to prevent potential issues for query completion. Before training, we prepend a start token to the instructions and append an end token to the descriptions. The training loss is computed only on the tokens of the descriptions and the end tokens, while excluding those of the instructions.

**Evalution Strategy.** For performance evaluation, we use the $80$ class names from `MS-COCO` dataset as query objectives. These include common objects such as trains, cars, and animals, as well as more specific categories like teddy bear, fire hydrant, and toothbrush. Based on the capitalization of each class name, we prepend either "a" or "an" to form the input queries. Since we focus on controlling the quality of retrieved images, we use aesthetic and relevance scores as the evaluation metrics. We calculate and report the average aesthetic and relevance scores of the retrieved images across all input queries as the final evaluation performance.

Table 4: Retrieval quality of various methods on `MS-COCO`, where **L** (Low), **M** (Medium), and **H** (High) indicate the quality conditions for retrieval, and Ctrl specifies whether the method enables controllable retrieval over image quality. For both average relevance (Ave Rel) and average aesthetics (Ave Aes), higher values indicate better retrieval quality.

| Quality | Aes Cond / Rel Cond | **L** / **L** | **L** / **M** | **L** / **H** | **M** / **L** | **M** / **M** | **M** / **H** | **H** / **L** | **H** / **M** | **H** / **H** | Ctrl ? |
|---|---|---|---|---|---|---|---|---|---|---|---|
| Prefix | Ave Aes | 4.817 | 4.817 | 4.817 | 4.817 | 4.817 | 4.817 | 4.817 | 4.817 | 4.817 | × |
| | Ave Rel | 0.349 | 0.349 | 0.349 | 0.349 | 0.349 | 0.349 | 0.349 | 0.349 | 0.349 | |
| LLaMA3 | Ave Aes | 4.903 | 4.891 | 4.855 | 4.916 | 4.875 | 4.880 | 4.871 | 4.858 | 4.911 | × |
| | Ave Rel | 0.348 | 0.349 | 0.347 | 0.348 | 0.349 | 0.347 | 0.348 | 0.350 | 0.344 | |
| GPT-4o | Ave Aes | 4.673 | 4.754 | 4.686 | 4.782 | 4.808 | 4.880 | 4.838 | 5.075 | 5.048 | × |
| | Ave Rel | 0.371 | 0.357 | 0.354 | 0.360 | 0.358 | 0.350 | 0.359 | 0.352 | 0.351 | |
| PT | Ave Aes | 4.819 | 4.793 | 4.789 | 4.829 | 4.810 | 4.828 | 4.794 | 4.826 | 4.820 | × |
| | Ave Rel | 0.343 | 0.340 | 0.344 | 0.348 | 0.339 | 0.344 | 0.346 | 0.343 | 0.340 | |
| FT | Ave Aes | 4.925 | 4.845 | 4.848 | 4.882 | 4.934 | 4.990 | 4.849 | 4.941 | 4.929 | × |
| | Ave Rel | 0.370 | 0.367 | 0.366 | 0.368 | 0.368 | 0.365 | 0.371 | 0.371 | 0.367 | |
| FT-CoCa | Ave Aes | 4.878 | 4.852 | 4.859 | 4.902 | 4.858 | 4.941 | 4.952 | 4.961 | 4.944 | × |
| | Ave Rel | 0.346 | 0.351 | 0.356 | 0.349 | 0.350 | 0.354 | 0.345 | 0.352 | 0.352 | |
| FT-Blip2 | Ave Aes | 4.828 | 4.815 | 4.785 | 4.932 | 4.894 | 4.893 | 5.034 | 4.948 | 4.933 | × |
| | Ave Rel | 0.350 | 0.352 | 0.356 | 0.344 | 0.351 | 0.353 | 0.345 | 0.351 | 0.347 | |
| Ours | Ave Aes | 4.811 | 4.790 | 4.773 | 4.911 | 4.873 | 4.862 | 5.016 | 4.983 | 5.024 | √ |
| | Ave Rel | 0.356 | 0.370 | 0.382 | 0.354 | 0.370 | 0.387 | 0.352 | 0.365 | 0.387 | |

## 4.2 QUALITATIVE VALIDATION

We first perform qualitative analysis to validate whether our approach effectively achieves quality control in retrieval. In Tables 1 and 2, we present three retrieved images per query, along with their corresponding completed queries and quality scores under three different quality conditions, including "low relevance, low aesthetic", "medium relevance, medium aesthetic", and "high relevance, high aesthetic". As shown, our method generates distinct query completions for different quality conditions. From left to right, as the quality level improves, both aesthetic and relevance scores increase accordingly. This demonstrates that our proposed method effectively controls the quality of the retrieved images. We provide more qualitative results with failure cases in the Appendix A.5.

## 4.3 QUANTITATIVE VALIDATION

Since there are no existing text-to-image retrieval methods that can be directly applicable to the proposed QCR task, we design the following baselines for quantitative comparison: a) *Prefix*: using original short queries directly without query completion; b) *PT (Pretrained)*: using a pretrained LLM for query completion without finetuning; c) *FT (Finetuned)*: finetuning a pretrained LLM on textual descriptions while conditioning on randomly generated quality scores; and d) *general-purpose LLMs*, including the LLaMA-3 (LLaMA-3-8B-Instruct) (Grattafiori et al., 2024) and GPT-4o (via official API) (OpenAI et al., 2024).

Tables 3 and 4 report the retrieval performance of the baseline models and our proposed method with Qwen2.5 (Yang et al., 2025) on the two datasets, respectively. From these tables, we make the following key observations. ① Prefix-only retrieval yields unsatisfactory quality performance, highlighting the necessity of query completion. ② Pretrained models (without finetuning) for query completion degrade retrieval quality, performing worse than using only the query prefix in most cases. This is because these pretrained models tend to generate irrelevant words, negatively impacting retrieval performance. ③ Finetuning on textual descriptions improves both relevance and aesthetics compared to prefix-only and pretrained models. However, models finetuned on randomly assigned scores fail to effectively control the quality of the retrieved images. ④ Our method not only enhances retrieval under high-quality conditions but also excels in quality control, demonstrating strong adaptability regardless of whether it is trained on real or generated captions.

Table 5: Results with five quality levels.

| $\mathcal{M}$ | | Relevance (Red → Red) | | | | |
|---|---|---|---|---|---|---|
| | | VL | L | M | H | VH |
| Aesthetics (Green ↓ Green) | VL | 4.597 | 4.507 | 4.610 | 4.529 | 4.445 |
| | | 0.355 | 0.364 | 0.375 | 0.382 | 0.397 |
| | L | 4.805 | 4.765 | 4.825 | 4.729 | 4.761 |
| | | 0.353 | 0.366 | 0.369 | 0.380 | 0.392 |
| | M | 4.909 | 4.961 | 4.878 | 4.889 | 4.901 |
| | | 0.355 | 0.3642 | 0.370 | 0.3754 | 0.390 |
| | H | 5.028 | 4.967 | 5.045 | 4.952 | 5.009 |
| | | 0.355 | 0.365 | 0.370 | 0.374 | 0.387 |
| | VH | 5.282 | 5.153 | 5.263 | 5.245 | 5.121 |
| | | 0.355 | 0.363 | 0.371 | 0.378 | 0.389 |

Table 6: Comparison with post-retrieval filtering, where the *rerank* method first retrieves the top-$k$ images based on relevance and then reorders the candidates by aesthetic scores to identify the best result.

| | $k$ | 1 | 2 | 3 | 5 | 10 |
|---|---|---|---|---|---|---|
| Rerank | Aes | 4.735 | 4.947 | 5.014 | 5.198 | 5.313 |
| | Rel | 0.350 | 0.348 | 0.347 | 0.345 | 0.341 |
| LLaMA3 | Aes | 4.842 | 5.071 | 5.154 | 5.298 | 5.377 |
| | Rel | 0.352 | 0.349 | 0.347 | 0.342 | 0.337 |
| GPT-4o | Aes | 5.056 | 5.205 | 5.293 | 5.393 | 5.518 |
| | Rel | 0.361 | 0.356 | 0.353 | 0.349 | 0.343 |
| Ours | Aes | 5.236 | 5.320 | 5.364 | 5.432 | 5.533 |
| | Rel | 0.387 | 0.385 | 0.381 | 0.376 | 0.366 |

## 4.4 DATASET DEPENDENCE

To achieve quality control in retrieval, the model should be tailored to the specific dataset, as different datasets exhibit varying quality characteristics. To illustrate this, we conduct cross-dataset retrieval experiments. Specifically, we evaluate retrieval quality on MS-COCO using queries completed by the model finetuned on Flickr2.4M. In Table 4, we assess FT-CoCa and FT-Blip2, which are finetuned on descriptions generated by CoCa and Blip2, respectively. The results indicate that both models achieve higher aesthetic scores as quality conditions improve, suggesting that aesthetically relevant semantic cues may be universal across natural images. Nevertheless, they consistently exhibit low relevance across all quality conditions. This limitation stems from the dataset mismatch between the query completion and image retrieval stages, since the two datasets encode different semantic information. See Appendix A.3 for additional analysis and results.

## 4.5 FURTHER VALIDATION

Table 5 presents the results on the Flickr2.4M dataset across five quality levels: VL (Very-Low), L (Low), M (Medium), H (High), and VH (Very-High). As shown, our method effectively enables fine-grained control over the quality of retrieved images, adhering to more nuanced descriptive constraints. We also compare against a post-retrieval filtering baseline that first retrieves images based on relevance and then re-ranks the results by aesthetic scores. The comparison results are listed Table 6. As shown, this two-stage strategy is unreliable for short queries, which typically offer vague representations and limited descriptive cues. As a result, the initial retrieval set tends to be semantically broad and aesthetically subpar, leaving little room for the re-ranking step to improve. While increasing $k$ can surface images with higher aesthetic quality, it typically comes at the cost of reduced semantic relevance, illustrating a trade-off between these two quality dimensions. In contrast, our method performs quality control during the query stage, which inherently guides retrieval toward the desired quality level. This quality-aware conditioning cannot be achieved by the two-step baseline, which lacks knowledge of the dataset's quality distribution and operates in a detached, post-hoc manner. See Appendix A.5 for more experimental results.

## 5 CONCLUSION

We presented a quality-controllable retrieval framework to address the limitations of short and underspecified text queries in text-to-image retrieval. Our key idea is to enrich queries using a generative language model conditioned on discretized quality levels, enabling retrieval that is both semantically expressive and aligned with user quality preferences. Extensive experiments demonstrate that our proposed QCQC approach effectively improves retrieval quality, serving as a flexible augmentation to existing VLMs while enabling quality control in retrieval. Future work will extend our method to other dimensions of quality beyond relevance and aesthetics, such as interestingness, diversity, or user personalization. We hope this work inspires further research on integrating controllable language-based query enrichment with large-scale multimodal retrieval systems.

## STATEMENTS

### ETHICS STATEMENT

This work investigates large language models for query completion in text-to-image retrieval, where image quality information is integrated into the training process. The study relies on publicly available datasets and does not involve human subjects, private information, or sensitive content. We acknowledge that retrieval models may inherit biases present in the underlying vision–language datasets; however, our approach does not introduce new data and instead focuses on methodological contributions. The models and results are intended solely for academic research, and no harmful or deceptive applications are pursued. We adhere to the ICLR Code of Ethics and confirm that this research complies with principles of fairness, transparency, and responsible use.

### REPRODUCIBILITY STATEMENT

We are committed to ensuring the reproducibility of our work. We have released our code at GitHub. Theoretical results are stated with all necessary assumptions in the main text and the complete proofs are provided in the Appendix. Experimental settings are included in the main body. Together, these resources are intended to enable full replication and verification of our results.

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

# A APPENDIX

## A.1 PROOFS

*Proof of Lemma 1.* Since $\mathrm{col}(\boldsymbol{X}_I) = \mathcal{U}$, we have $\boldsymbol{I}_r = \boldsymbol{X}_I \boldsymbol{X}_I^\dagger$. Hence

$$\boldsymbol{X}_I + \boldsymbol{P}\boldsymbol{Y}_I = \boldsymbol{X}_I + \boldsymbol{X}_I \boldsymbol{X}_I^\dagger \boldsymbol{P}\boldsymbol{Y}_I = \boldsymbol{X}_I(\boldsymbol{I}_r + \boldsymbol{X}_I^\dagger \boldsymbol{P}\boldsymbol{Y}_I). \tag{6}$$

This factorization assumes $\boldsymbol{P}\boldsymbol{Y}_I$ lies in $\mathrm{col}(\boldsymbol{X}_I)$, which is true because $\boldsymbol{P}$ projects onto $\mathrm{col}(\boldsymbol{X})$ and $\mathrm{col}(\boldsymbol{X}) = \mathrm{col}(\boldsymbol{X}_I)$. Considering the Neumann Series argument: for any matrix $\boldsymbol{E}$, if $\|\boldsymbol{E}\|_2 < 1$, then $(\boldsymbol{I} + \boldsymbol{E})$ is invertible (i.e., full rank). Given $\|\boldsymbol{X}_I^\dagger \boldsymbol{P}\boldsymbol{Y}_I\|_2 < 1$, $(\boldsymbol{I}_r + \boldsymbol{X}_I^\dagger \boldsymbol{P}\boldsymbol{Y}_I)$ is invertible. Thus, $\mathrm{rank}(\boldsymbol{X}_I + \boldsymbol{P}\boldsymbol{Y}_I) = \mathrm{rank}(\boldsymbol{X}_I) = r$. $\qquad\square$

*Proof of Proposition 1.* Since right-multiplication by the orthogonal matrix $\boldsymbol{V} = [\boldsymbol{V}_S\ \boldsymbol{V}_\perp]$ is rank-preserving, we analyze the following matrices:

$$\boldsymbol{A}' := \boldsymbol{A}\boldsymbol{V} = \boldsymbol{A}[\boldsymbol{V}_S\ \boldsymbol{V}_\perp] = [\boldsymbol{A}\boldsymbol{V}_S\ \boldsymbol{A}\boldsymbol{V}_\perp] = [\boldsymbol{A}_S\ \boldsymbol{0}], \tag{7}$$

$$\boldsymbol{\Delta}' := \boldsymbol{\Delta}\boldsymbol{V} = \boldsymbol{\Delta}[\boldsymbol{V}_S\ \boldsymbol{V}_\perp] = [\boldsymbol{\Delta}\boldsymbol{V}_S\ \boldsymbol{\Delta}\boldsymbol{V}_\perp] = [\boldsymbol{\Delta}_S\ \boldsymbol{\Delta}_\perp], \tag{8}$$

$$\boldsymbol{B}' := \boldsymbol{B}\boldsymbol{V} = (\boldsymbol{A} + \boldsymbol{\Delta})\boldsymbol{V} = \boldsymbol{A}\boldsymbol{V} + \boldsymbol{\Delta}\boldsymbol{V} = \boldsymbol{A}' + \boldsymbol{\Delta}' = [\boldsymbol{A}_S + \boldsymbol{\Delta}_S\ \boldsymbol{\Delta}_\perp], \tag{9}$$

$$\boldsymbol{C}' := \boldsymbol{C}\boldsymbol{V} = \boldsymbol{C}[\boldsymbol{V}_S\ \boldsymbol{V}_\perp] = [\boldsymbol{C}\boldsymbol{V}_S\ \boldsymbol{C}\boldsymbol{V}_\perp] = [\boldsymbol{C}_S\ \boldsymbol{C}_\perp]. \tag{10}$$

Then, for the score matrices, we have:

$$\boldsymbol{S}_A = \boldsymbol{A}\boldsymbol{C}^\top = (\boldsymbol{A}\boldsymbol{V})(\boldsymbol{C}\boldsymbol{V})^\top = \boldsymbol{A}'\boldsymbol{C}'^\top = [\boldsymbol{A}_S\ \boldsymbol{0}]\begin{bmatrix} \boldsymbol{C}_S^\top \\ \boldsymbol{C}_\perp^\top \end{bmatrix} = \boldsymbol{A}_S\boldsymbol{C}_S^\top, \tag{11}$$

$$\boldsymbol{S}_B = \boldsymbol{B}\boldsymbol{C}^\top = \boldsymbol{B}'\boldsymbol{C}'^\top = [\boldsymbol{A}_S + \boldsymbol{\Delta}_S\ \boldsymbol{\Delta}_\perp]\begin{bmatrix} \boldsymbol{C}_S^\top \\ \boldsymbol{C}_\perp^\top \end{bmatrix} = \underbrace{(\boldsymbol{A}_S + \boldsymbol{\Delta}_S)\boldsymbol{C}_S^\top}_{\boldsymbol{X}} + \underbrace{\boldsymbol{\Delta}_\perp\boldsymbol{C}_\perp^\top}_{\boldsymbol{Y}}. \tag{12}$$

By the SVD construction, $\boldsymbol{A}_S$ has full column rank $r$ and $\sigma_{\min}(\boldsymbol{A}_S) = \sigma_r(\boldsymbol{A}) > 0$. Since $\boldsymbol{\Delta}_S = \boldsymbol{\Delta}\boldsymbol{V}_S$ and $\boldsymbol{V}_S$ is orthogonal (i.e., $\|\boldsymbol{V}_S\|_2 = 1$), it follows that

$$\|\boldsymbol{\Delta}_S\|_2 = \|\boldsymbol{\Delta}\boldsymbol{V}_S\|_2 \leq \|\boldsymbol{\Delta}\|_2. \tag{13}$$

According to assumption (i), we know $\|\boldsymbol{\Delta}_S\|_2 \leq \|\boldsymbol{\Delta}\|_2 < \sigma_r(\boldsymbol{A}) = \sigma_{\min}(\boldsymbol{A}_S)$. The standard minimum-singular-value perturbation argument (or Weyl's inequality in spectral norm form) yields that $\boldsymbol{A}_S + \boldsymbol{\Delta}_S$ remains full column rank $r$. Since left multiplication by a full-column-rank matrix does not change rank, it follows that:

$$\mathrm{rank}(\boldsymbol{X}) = \mathrm{rank}\big((\boldsymbol{A}_S + \boldsymbol{\Delta}_S)\boldsymbol{C}_S^\top\big) = \mathrm{rank}(\boldsymbol{C}_S^\top) = \mathrm{rank}(\boldsymbol{C}_S), \tag{14}$$

$$\mathrm{rank}(\boldsymbol{S}_A) = \mathrm{rank}(\boldsymbol{A}_S\boldsymbol{C}_S^\top) = \mathrm{rank}(\boldsymbol{C}_S) = \mathrm{rank}(\boldsymbol{X}). \tag{15}$$

According to the assumption (ii), there exists a set $I$ of size $r$ such that $\boldsymbol{X}_I$ is a basis for $\mathrm{col}(\boldsymbol{X})$. Since $|\boldsymbol{X}_I| = r$, the dimension of the column space must be $r$. As a result,

$$\mathrm{rank}(\boldsymbol{S}_A) = \mathrm{rank}(\boldsymbol{X}) = r. \tag{16}$$

To prove $\mathrm{rank}(\boldsymbol{S}_B) \geq \mathrm{rank}(\boldsymbol{S}_A)$, we construct a linear operator (matrix) $\boldsymbol{T}$ that decouples the column space of $\boldsymbol{X}$ from the rest of the space. Let $\boldsymbol{P} := \boldsymbol{P}_X$ be the orthogonal projector onto $\mathcal{U} := \mathrm{col}(\boldsymbol{X})$, $\boldsymbol{Z} := (\boldsymbol{I} - \boldsymbol{P})\boldsymbol{Y}$ be the component of $\boldsymbol{Y}$ orthogonal to $\boldsymbol{X}$, and $\boldsymbol{P}_{\boldsymbol{Z}_I}$ be the orthogonal projector onto $\mathrm{col}(\boldsymbol{Z}_I)$. We consider the linear operator:

$$\boldsymbol{T} = \begin{bmatrix} \boldsymbol{P} \\ (\boldsymbol{I} - \boldsymbol{P}_{\boldsymbol{Z}_I})(\boldsymbol{I} - \boldsymbol{P}) \end{bmatrix}. \tag{17}$$

Since $\boldsymbol{P}$ projects onto $\mathrm{col}(\boldsymbol{X})$, $\boldsymbol{P}\boldsymbol{X} = \boldsymbol{X}$, and $(\boldsymbol{I} - \boldsymbol{P})\boldsymbol{X} = \boldsymbol{0}$. Given $\boldsymbol{Z} := (\boldsymbol{I} - \boldsymbol{P})\boldsymbol{Y}$, we have

$$\boldsymbol{T}\boldsymbol{S}_B = \boldsymbol{T}(\boldsymbol{X} + \boldsymbol{Y}) = \begin{bmatrix} \boldsymbol{P}\boldsymbol{X} + \boldsymbol{P}\boldsymbol{Y} \\ (\boldsymbol{I} - \boldsymbol{P}_{\boldsymbol{Z}_I})(\boldsymbol{I} - \boldsymbol{P})(\boldsymbol{X} + \boldsymbol{Y}) \end{bmatrix} = \begin{bmatrix} \boldsymbol{X} + \boldsymbol{P}\boldsymbol{Y} \\ (\boldsymbol{I} - \boldsymbol{P}_{\boldsymbol{Z}_I})\boldsymbol{Z} \end{bmatrix}. \tag{18}$$

Restricting to $I \cup K$, we analyze the submatrix formed by columns indexed by $I$ (size $r$) and $K$ (size determined by $k$), which are disjoint:

$$(\boldsymbol{TS}_B)_{:,I\cup K} = \begin{bmatrix} \boldsymbol{X}_I + \boldsymbol{PY}_I & \boldsymbol{X}_K + \boldsymbol{PY}_K \\ (\boldsymbol{I} - \boldsymbol{P}_{\boldsymbol{Z}_I})\boldsymbol{Z}_I & (\boldsymbol{I} - \boldsymbol{P}_{\boldsymbol{Z}_I})\boldsymbol{Z}_K \end{bmatrix} = \begin{bmatrix} \boldsymbol{X}_I + \boldsymbol{PY}_I & \boldsymbol{X}_K + \boldsymbol{PY}_K \\ \boldsymbol{0} & (\boldsymbol{I} - \boldsymbol{P}_{\boldsymbol{Z}_I})\boldsymbol{Z}_K \end{bmatrix}. \quad (19)$$

Since $\boldsymbol{P}_{\boldsymbol{Z}_I}$ projects onto the column space of $\boldsymbol{Z}_I$, the residual $(\boldsymbol{I} - \boldsymbol{P}_{\boldsymbol{Z}_I})\boldsymbol{Z}_I$ is exactly zero. Because the matrix becomes block upper-triangular, its rank is the sum of the ranks of the diagonal blocks.

By Lemma 1, the top-left block has rank $r$. By assumption (iv), the bottom-right block has rank $k \geq 1$. Thus block-triangular rank additivity yields

$$\mathrm{rank}\big((\boldsymbol{TS}_B)_{:,I\cup K}\big) = r + k. \quad (20)$$

Since the rank of a submatrix (columns $I \cup K$) is a lower bound for the rank of the full matrix, and left multiplication by $\boldsymbol{T}$ cannot increase the rank of the original matrix, we have the inequality:

$$\mathrm{rank}(\boldsymbol{S}_B) \geq \mathrm{rank}(\boldsymbol{TS}_B) \geq \mathrm{rank}\big((\boldsymbol{TS}_B)_{:,I\cup K}\big). \quad (21)$$

Given $k \geq 1$ and $\mathrm{rank}(\boldsymbol{S}_A) = r$, we have:

$$\mathrm{rank}(\boldsymbol{S}_B) \geq r + k > r = \mathrm{rank}(\boldsymbol{S}_A). \quad (22)$$

Here, we complete the proof. $\square$

*Remark.* We decompose $\boldsymbol{\Delta}$ into two parts: one $(\boldsymbol{\Delta}_S)$ that lies in the original row space of $\boldsymbol{A}$, and another $(\boldsymbol{\Delta}_\perp)$ that introduces directions outside this space. Assumption (i) ensures the in-span perturbation $\boldsymbol{\Delta}_S$ is not too large (controlled by $\sigma_r(\boldsymbol{A})$) so the original $r$ query directions in $\boldsymbol{A}$ are not destroyed by completion. Assumption (ii) asserts that we can select $r$ columns from $\boldsymbol{X}$ that span $\mathcal{U}$. This fixes a stable $r$-dimensional basis for the existing subspace. Assumption (iii) claims that adding the projected perturbation $\boldsymbol{PY}_I$ does not reduce the independence of these $r$ columns. Thus the original $r$-dimensional structure is preserved. Assumption (iv) requires that there exist $k \geq 1$ columns outside $I$ whose orthogonal components (after removing projections onto both $\mathcal{U}$ and $\mathrm{col}(\boldsymbol{Z}_I)$) are linearly independent. These contribute $k$ genuinely new directions in $\mathcal{U}^\perp$. Together, these assumptions ensure that the rank of $\boldsymbol{S}_B$ contains at least the $r$ preserved dimensions from $\mathcal{U}$ plus the $k$ fresh orthogonal ones. Consequently, $\boldsymbol{S}_B$ can express more independent scoring patterns and has the ability to potentially make finer-grained distinctions.

## A.2 Related Work

### A.2.1 Large Language Models

Large language models (LLMs) are a class of foundation models designed to process, understand, and generate natural language at scale. With fine-tuning and prompting, these models excel across a variety of tasks, including text generation, summarization, reasoning, translation, and coding (Devlin et al., 2019; Radford et al., 2019; Brown et al., 2020). Notable examples, such as LLaMA3 (Grattafiori et al., 2024), GPT-4o (OpenAI et al., 2024), and Qwen2.5 (Yang et al., 2025), contain billions of parameters and are trained on extensive textual datasets. The large-scale pretraining enables them to capture complex contextual, semantic, and syntactic relationships in natural language. In this work, we utilize pretrained LLMs for query modification to tackle the proposed QCR task. By integrating quality information as conditions, our model autonomously learns to generate quality-aware details for query extension. This provides users with multiple visible query suggestions, allowing them to explore diverse retrieval results.

### A.2.2 Vision-Language Models

Vision-language models (VLMs) have become the de facto foundation for image-text tasks, demonstrating exceptional potential across a variety of applications (Maniparambil et al., 2024; Dong et al., 2025; Huang et al., 2026). Pioneering work such as CLIP (Radford et al., 2021) and ALIGN (Jia et al., 2021) learn directly from raw texts about images by aligning them in a shared embedding space. CoCa (Yu et al., 2022) combines contrastive loss with captioning loss to train an image-text encoder-decoder model, effectively integrating capabilities from both contrastive and generative approaches. Blip2 (Li et al., 2023) bridges the modality gap with a lightweight Q-Former to improve

pretraining efficiency. In this paper, we adopt joint-embedding VLMs like CLIP as foundation models for text-to-image retrieval. Instead of fine-tuning the VLMs on target datasets, we keep them frozen and focus on refining textual queries to achieve both quality improvement and control over the retrieved images. Improving existing VLMs for retrieval quality control is orthogonal to our approach and represents a promising direction for our future research.

### A.2.3 TEXT-TO-IMAGE RETRIEVAL

Text-to-image retrieval aims to identify the most relevant images from a database given a natural language query (Chen et al., 2021; Lu et al., 2022; Pacini et al., 2024; Wu et al., 2025). It plays a critical role in applications such as visual search, e-commerce, and content-based recommendation. Recent advances in VLMs (Radford et al., 2021; Gao et al., 2022; Yu et al., 2022; Jia et al., 2021) have significantly improved performance on this task by learning powerful cross-modal representations. These models map images and texts into a shared embedding space, typically through contrastive learning on web-scaled image-text pairs. However, existing retrieval systems are primarily optimized to return the top-k images that are semantically aligned with the input query. They overlook other crucial dimensions that strongly affect user satisfaction in practical scenarios, such as aesthetic appeal (Yi et al., 2023), interestingness (Gygli et al., 2013; Abdullahu & Grabner, 2024), or low-level visual quality (Wang et al., 2021; 2023a; Zeng et al., 2025). In this work, we advocate for incorporating quality control into retrieval. By allowing users to explicitly influence the quality attributes of the returned results, we enable a more personalized and controllable search experience, moving beyond simple semantic matching toward a more adaptive, user-centric paradigm.

### A.2.4 QUERY COMPLETION

Query completion (QC) aims to extend user short inputs, referred to as *query prefixes*, by generating longer and more informative *query completions*. It is a widely used technique that helps users better articulate their intent and resolve potential query ambiguity. Traditional QC methods rely on factors such as user profiles, query libraries, and prior search history to extend prefixes into query completions, which limits their applicability to unforeseen prefixes (Bar-Yossef & Kraus, 2011; Mitra & Craswell, 2015; Cai & De Rijke, 2016). Recently, several generative approaches have been proposed for query completion with arbitrary prefixes, primarily for text generation and document retrieval tasks (Lee et al., 2021; Wang et al., 2023b; Lei et al., 2024). QC-based methods for text-to-image retrieval remain scarce, with only a few related works. Zhu et al. (2024) enhance interactive image retrieval through query rewriting based on user relevance feedback, while Sun et al. (2024) leverage LLMs to generate product-aware query completions. Pacini et al. (2024) refine text queries using visual feedback to improve text-to-image retrieval performance. However, these approaches primarily focus on query suggestion rather than achieving control over the quality of retrieved images. In contrast, we tailor query completion to enhance retrieval quality, making the first attempt to adapt it to a given search corpus for quality-controllable retrieval.

### A.3 ANALYSIS OF DATASET DEPENDENCY

Our work focuses on text-to-image retrieval, where the goal is to retrieve relevant images from a fixed dataset based on a textual query. This task is inherently dataset-dependent, as the retrieval process relies entirely on the available images within the dataset. Therefore, the query is crucial in this task: the more specific and detailed the query, the easier the retrieval system can match it to the corresponding image. Conversely, short or vague queries make it significantly more difficult for the system to identify the intended image. That is why our proposed QCQC method aims to enrich the original short queries with more specific, quality-aware details. It it notable that these details are not randomly generated. Instead, they are learned directly from the dataset itself by fine-tuning the LLM to fit the captions. As a result, the completed queries remain dataset-dependent and contextually relevant. The additional details are not unnecessary, as they provide essential guidance to the retrieval system, making it accurately identify images with desired quality.

### A.4 ANALYSIS OF SCORE DIFFERENCES

In Tables 3 and 4, the average quality scores across low, medium, and high conditions may appear close for a set of short queries. This is expected due to dataset limitations. As shown in Figure

Table 7: Retrieval quality with five quality levels on CoCa.

| $\mathcal{M}$ | | Relevance (Red → Red) | | | | |
| --- | --- | --- | --- | --- | --- | --- |
| | | VL | L | M | H | VH |
| Aesthetics (Green ← Green) | VL | 4.581 
 0.355 | 4.551 
 0.364 | 4.559 
 0.372 | 4.579 
 0.376 | 4.507 
 0.382 |
| | L | 4.870 
 0.357 | 4.792 
 0.363 | 4.784 
 0.370 | 4.748 
 0.376 | 4.718 
 0.383 |
| | M | 4.882 
 0.356 | 4.954 
 0.366 | 4.863 
 0.371 | 4.849 
 0.377 | 4.820 
 0.381 |
| | H | 5.054 
 0.355 | 5.048 
 0.362 | 5.005 
 0.371 | 5.019 
 0.370 | 4.998 
 0.381 |
| | VH | 5.159 
 0.352 | 5.166 
 0.366 | 5.161 
 0.369 | 5.135 
 0.373 | 5.084 
 0.386 |

1, the relevance scores across both Flickr2.4M and MS-COCO are not uniformly distributed, and images with extremely low or high scores are rare. For instance, on Flickr2.4M, the relevance scores range from $0.252$ to $0.562$, and the entire span across the whole dataset is only about $0.3$ (where the extreme values correspond to two images from different classes). When retrieving images for a single query, the available results often fall within a narrower score range (much smaller than $0.3$) because the dataset lacks images at both ends of the quality spectrum (sparsely distributed). For example, if all images retrieved from the query "a dog" have aesthetic scores between $3.8$ and $4.7$ (due to dataset limitations), even under the "High" condition, the best available image might score $4.7$ which lies in a low range (given the whole range : $[2.782, 6.961]$). But it is still higher than the score of $3.8$ under the "Low" condition. Thus, the method is still effectively ranking and retrieving better images within the constraints of the dataset.

Despite this dataset-level constraint that limits the score differences, our method demonstrates effective ranking ability and a consistent, meaningful trend. As shown from left to right in Tables 1 and 2, both the retrieved image scores and their visual appeal improve progressively as the quality condition increases. This pattern is further supported by quantitative results in Tables 3 and 4, where the average quality scores clearly increase across the low, medium, and high conditions. This behavior cannot be reproduced by baseline methods that lack quality consideration in retrieval.

## A.5 ADDITIONAL EXPERIMENTAL RESULTS

In Table 7, we present the results on the `Flickr2.4M` dataset across five quality levels using CoCa as the caption model. Tables 8 and 9 presents the quantitative results of our method using GPT2 (Radford et al., 2019) as the backbone. In addition, we provide more qualitative comparison on the two datasets in Tables 10-12. These results further demonstrate that the proposed QCQC method enables controllable retrieval quality. In rare cases, the completed queries may not align with the semantics of the query prefixes. This occurs when the query completion model generates a sentence referencing different objects. Refer to Table 13 for examples of such cases.

## A.6 LIMITATION

The relevance and aesthetic quality of the retrieved images depend on the reliability of the VLMs and aesthetic evaluation models. If these models are not sufficiently reliable, retrieval performance can be significantly affected. In addition, the model needs to perceive image quality within the datasets to achieve quality control in retrieval. However, the retrieval datasets may sometimes lack the granularity needed to differentiate between high-quality and low-quality images. In some instances, the retrieval database may not contain high-quality or low-quality images that match specific queries.

## A.7 THE USE OF LARGE LANGUAGE MODELS (LLMS)

In preparing this paper, large language models were used as writing assistants for grammar checking and minor sentence rephrasing. All technical aspects of the work, including the design, implemen-

Table 8: Retrieval quality of various methods on `Flickr2.4M`. CoCa and Blip2 are used to generate textual descriptions; **L** (Low), **M** (Medium), and **H** (High) indicate the quality conditions; and Ctrl specifies whether the method enables controllable retrieval over quality. For both average relevance (Ave Rel) and average aesthetics (Ave Aes), higher values indicate better retrieval quality.

| Quality | VLM | Aes Cond / Rel Cond | L / L | L / M | L / H | M / L | M / M | M / H | H / L | H / M | H / H | Ctrl ? |
|---|---|---|---|---|---|---|---|---|---|---|---|---|
| Prefix | –– | Ave Aes | 4.735 | 4.735 | 4.735 | 4.735 | 4.735 | 4.735 | 4.735 | 4.735 | 4.735 | × |
|  |  | Ave Rel | 0.350 | 0.350 | 0.350 | 0.350 | 0.350 | 0.350 | 0.350 | 0.350 | 0.350 |  |
| LLaMA3 | –– | Ave Aes | 4.730 | 4.822 | 4.831 | 4.823 | 4.837 | 4.784 | 4.798 | 4.722 | 4.842 | × |
|  |  | Ave Rel | 0.351 | 0.351 | 0.351 | 0.353 | 0.351 | 0.350 | 0.354 | 0.354 | 0.352 |  |
| GPT-4o | –– | Ave Aes | 4.359 | 4.651 | 4.728 | 4.712 | 4.668 | 4.791 | 4.791 | 4.816 | 5.056 | × |
|  |  | Ave Rel | 0.378 | 0.361 | 0.357 | 0.358 | 0.360 | 0.356 | 0.361 | 0.357 | 0.361 |  |
| PT | –– | Ave Aes | 4.681 | 4.639 | 4.673 | 4.688 | 4.504 | 4.654 | 4.610 | 4.556 | 4.692 | × |
|  |  | Ave Rel | 0.351 | 0.344 | 0.350 | 0.350 | 0.346 | 0.347 | 0.349 | 0.352 | 0.352 |  |
| FT | CoCa | Ave Aes | 4.848 | 4.818 | 4.864 | 4.847 | 4.827 | 4.876 | 4.829 | 4.896 | 4.853 | × |
|  |  | Ave Rel | 0.366 | 0.365 | 0.367 | 0.365 | 0.363 | 0.366 | 0.367 | 0.366 | 0.368 |  |
| Ours | CoCa | Ave Aes | 4.646 | 4.674 | 4.632 | 4.878 | 4.921 | 4.894 | 5.182 | 5.095 | 5.124 | √ |
|  |  | Ave Rel | 0.354 | 0.372 | 0.382 | 0.355 | 0.369 | 0.386 | 0.357 | 0.366 | 0.385 |  |
| FT | Blip2 | Ave Aes | 4.838 | 4.674 | 4.744 | 4.592 | 4.599 | 4.772 | 4.727 | 4.749 | 4.818 | × |
|  |  | Ave Rel | 0.369 | 0.360 | 0.369 | 0.365 | 0.362 | 0.365 | 0.373 | 0.359 | 0.368 |  |
| Ours | Blip2 | Ave Aes | 4.528 | 4.560 | 4.470 | 4.948 | 4.946 | 4.885 | 5.266 | 5.160 | 5.236 | √ |
|  |  | Ave Rel | 0.355 | 0.374 | 0.393 | 0.354 | 0.374 | 0.391 | 0.354 | 0.367 | 0.387 |  |

Table 9: Retrieval quality of various methods on `MS-COCO`, where **L** (Low), **M** (Medium), and **H** (High) indicate the quality conditions for retrieval, and Ctrl specifies whether the method enables controllable retrieval over image quality. For both average relevance (Ave Rel) and average aesthetics (Ave Aes), higher values indicate better retrieval quality.

| Quality | Aes Cond / Rel Cond | L / L | L / M | L / H | M / L | M / M | M / H | H / L | H / M | H / H | Ctrl ? |
|---|---|---|---|---|---|---|---|---|---|---|---|
| Prefix | Ave Aes | 4.817 | 4.817 | 4.817 | 4.817 | 4.817 | 4.817 | 4.817 | 4.817 | 4.817 | × |
|  | Ave Rel | 0.349 | 0.349 | 0.349 | 0.349 | 0.349 | 0.349 | 0.349 | 0.349 | 0.349 |  |
| LLaMA3 | Ave Aes | 4.903 | 4.891 | 4.855 | 4.916 | 4.875 | 4.880 | 4.871 | 4.858 | 4.911 | × |
|  | Ave Rel | 0.348 | 0.349 | 0.347 | 0.348 | 0.349 | 0.347 | 0.348 | 0.350 | 0.344 |  |
| GPT-4o | Ave Aes | 4.673 | 4.754 | 4.686 | 4.782 | 4.808 | 4.880 | 4.838 | 5.075 | 5.048 | × |
|  | Ave Rel | 0.371 | 0.357 | 0.354 | 0.360 | 0.358 | 0.350 | 0.359 | 0.352 | 0.351 |  |
| PT | Ave Aes | 4.742 | 4.731 | 4.855 | 4.821 | 4.775 | 4.854 | 4.830 | 4.726 | 4.847 | × |
|  | Ave Rel | 0.347 | 0.345 | 0.350 | 0.349 | 0.344 | 0.345 | 0.351 | 0.347 | 0.344 |  |
| FT | Ave Aes | 4.785 | 4.820 | 4.866 | 4.813 | 4.852 | 4.888 | 4.833 | 4.919 | 4.960 | × |
|  | Ave Rel | 0.369 | 0.369 | 0.373 | 0.373 | 0.369 | 0.373 | 0.367 | 0.376 | 0.372 |  |
| FT-CoCa | Ave Aes | 4.890 | 4.889 | 4.793 | 4.885 | 4.939 | 4.903 | 4.950 | 5.004 | 4.898 | × |
|  | Ave Rel | 0.347 | 0.348 | 0.356 | 0.346 | 0.349 | 0.352 | 0.347 | 0.349 | 0.351 |  |
| FT-Blip2 | Ave Aes | 4.776 | 4.883 | 4.824 | 4.914 | 4.968 | 4.873 | 5.039 | 4.967 | 5.053 | × |
|  | Ave Rel | 0.349 | 0.351 | 0.352 | 0.344 | 0.349 | 0.350 | 0.343 | 0.349 | 0.349 |  |
| Ours | Ave Aes | 4.896 | 4.809 | 4.719 | 4.973 | 4.879 | 4.916 | 5.017 | 5.020 | 5.109 | √ |
|  | Ave Rel | 0.354 | 0.365 | 0.385 | 0.356 | 0.368 | 0.387 | 0.353 | 0.368 | 0.391 |  |

tation, and verification of experiments and analyses, were carried out by the authors. In our experiments, we include the LLaMA-3 (Grattafiori et al., 2024) and GPT-4o as query completion models (OpenAI et al., 2024) for empirical comparison.

Table 10: Query completions with their retrieved images and quality scores on MS-COCO

| Rel: Low, Aes: Low | Rel: Medium, Aes: Medium | Rel: High, Aes: High |
|---|---|---|
| 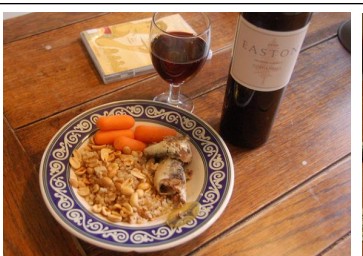 | 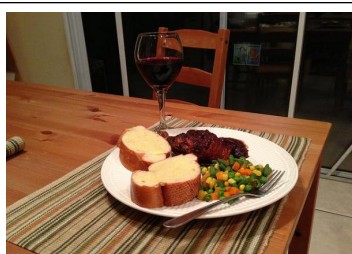 | 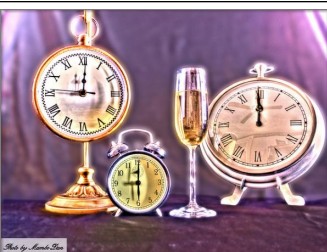 |
| ***a wine glass*** *next to a plate with some fish and veggies on it* 
 Aes 4.790, Rel 0.350 | ***a wine glass*** *next to a plate with some meat and vegetables on it* 
 Aes 5.209, Rel 0.376 | ***a wine glass*** *and three clocks all set at different times* 
 Aes 5.555, Rel 0.417 |
| 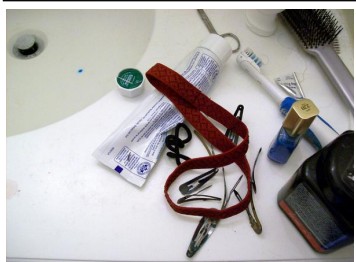 | 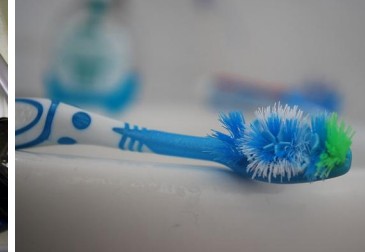 | 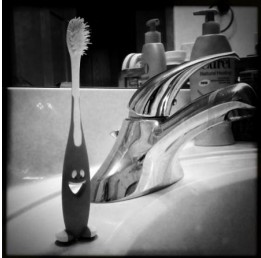 |
| ***a toothbrush*** *on a table with a bunch of scissors* 
 Aes 4.149, Rel 0.345 | ***a toothbrush*** *that is on down on the counter* 
 Aes 4.837, Rel 0.370 | ***a toothbrush*** *with a smiley face sitting on a sink* 
 Aes 5.184, Rel 0.412 |
| 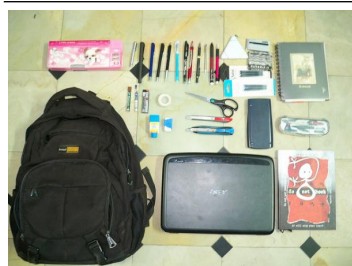 | 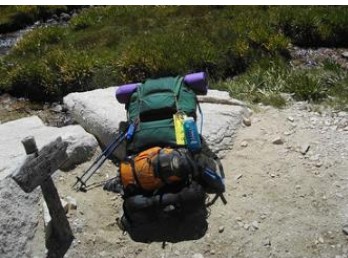 | 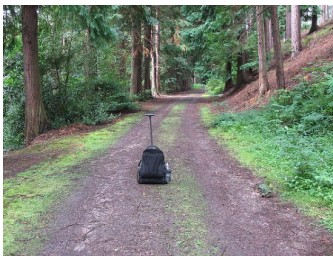 |
| ***a backpack*** *and a line of supplies laying out* 
 Aes 3.937, Rel 0.355 | ***a backpack*** *some water rocks and plants* 
 Aes 5.130, Rel 0.374 | ***a backpack*** *with rollers is sitting unattended in the middle of this forested dirt road* 
 Aes 5.231, Rel 0.470 |

Table 11: Query completions with their retrieved images and quality scores on `Flickr2.4M`

| **Rel: Low, Aes: Low** | **Rel: Medium, Aes: Medium** | **Rel: High, Aes: High** |
|---|---|---|
| 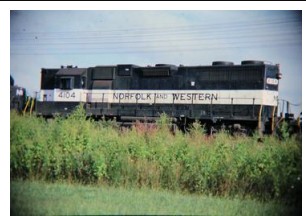 | 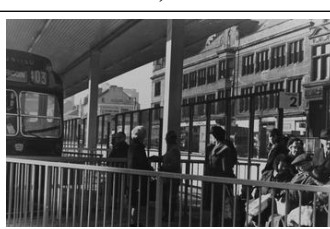 | 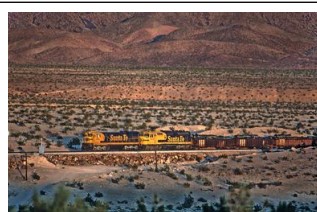 |
| ***a train*** *on a track next to a grassy field* 
 Aes 4.585, Rel 0.369 | ***a train*** *station with people waiting to board a bus* 
 Aes 4.910, Rel 0.380 | ***a train*** *in the desert* 

 Aes 5.488, Rel 0.391 |
| 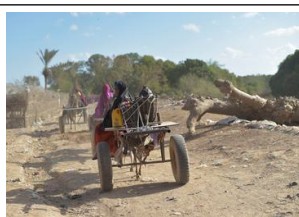 | 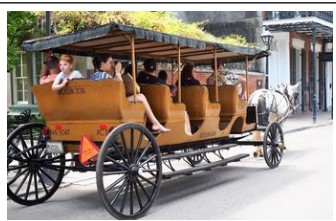 | 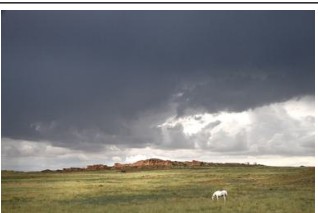 |
| ***a horse*** *drawn carriage on a dirt road* 
 Aes 4.718, Rel 0.357 | ***a horse*** *drawn carriage with people on it* 
 Aes 5.023, Rel 0.3773 | ***a horse*** *is grazing in a field under a cloudy sky* 
 Aes 5.207, Rel 0.390 |
| 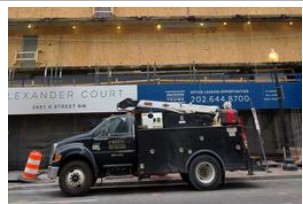 | 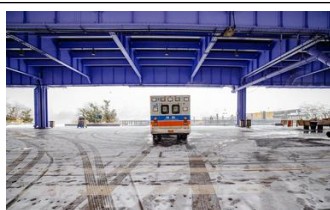 | 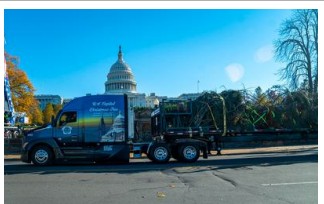 |
| ***a truck*** *is parked in front of a building* 
 Aes 4.800, Rel 0.331 | ***a truck*** *is parked under a bridge* 
 Aes 5.070, Rel 0.370 | ***a truck*** *is parked in front of the washington monument* 
 Aes 5.335, Rel 0.389 |
| 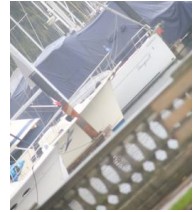 | 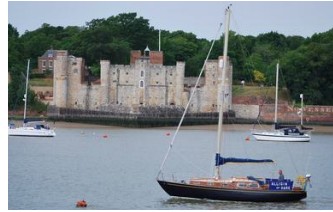 | 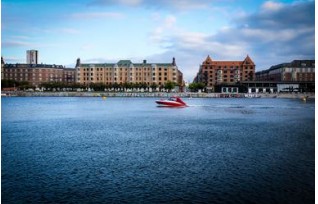 |
| ***a boat*** *docked in the water next to other boats* 
 Aes 3.814, Rel 0.358 | ***a boat*** *is in the water near a castle* 
 Aes 4.839, Rel 0.371 | ***a boat*** *on the water with buildings in the background* 
 Aes 5.113, Rel 0.396 |

Table 12: Query completions with their retrieved images and quality scores on `MS-COCO`

| Rel: Low, Aes: Low | Rel: Medium, Aes: Medium | Rel: High, Aes: High |
|---|---|---|
| 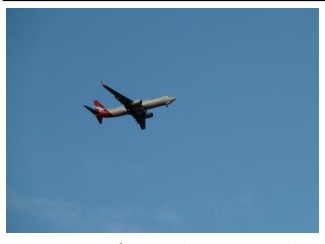 |  | 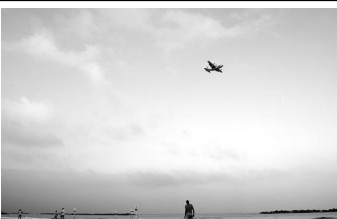 |
| ***an aeroplane*** *flying in the air with a big blue sky behind it* 
 Aes 4.536, Rel 0.354 | ***an aeroplane*** *flying high on a clear sky* 
 Aes 4.739, Rel 0.360 | **Query**: ***an aeroplane*** *flying over the beach and two guys standing on it* 
 Aes 5.516, Rel 0.425 |
| 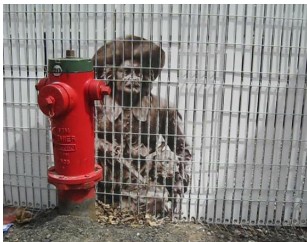 | 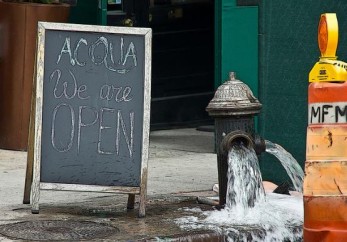 | 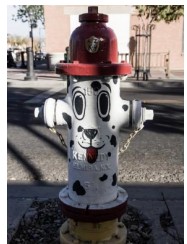 |
| ***a fire hydrant*** *stands in front of a bald eagle wall mural* 
 Aes 4.991, Rel 0.355 | ***a fire hydrant*** *sitting in front of a sign for a cafe* 
 Aes 5.511, Rel 0.368 | ***a fire hydrant*** *is painted to look like a dalmation* 
 Aes 5.799, Rel 0.447 |
| 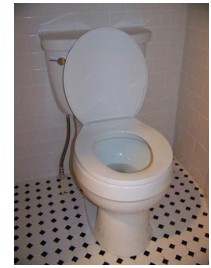 | 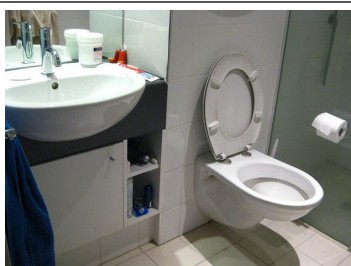 | 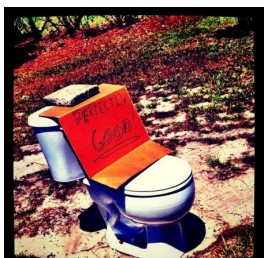 |
| ***a toilet*** *with a raised lid in some lavatory* 
 Aes 4.457, Rel 0.361 | ***a toilet*** *and sink in a small bathroom with a seat up* 
 Aes 4.502, Rel 0.372 | ***a toilet*** *is sitting outside with a sign on it* 
 Aes 5.264, Rel 0.394 |
| 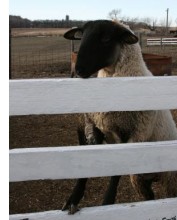 | 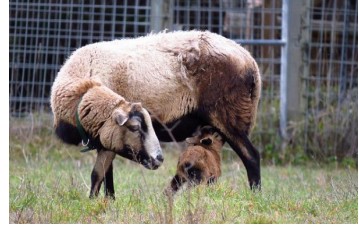 | 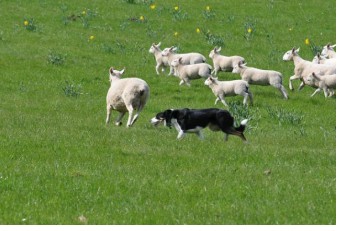 |
| ***a sheep*** *is standing on a white fence* 
 Aes 4.557, Rel 0.358 | ***a sheep*** *and baby sheep standing in a field* 
 Aes 5.014, Rel 0.378 | ***a sheep*** *dog herding sheep through a grass field* 
 Aes 5.213, Rel 0.388 |

Table 13: Some bad retrieval cases on the two datasets

| **Rel: Low, Aes: Low** | **Rel: Medium, Aes: Medium** | **Rel: High, Aes: High** |
|---|---|---|
| 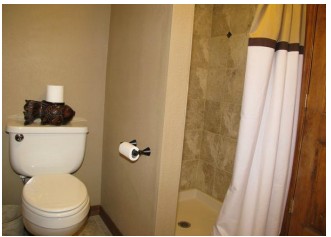 | 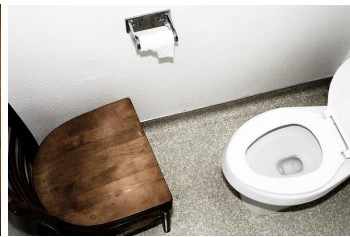 | 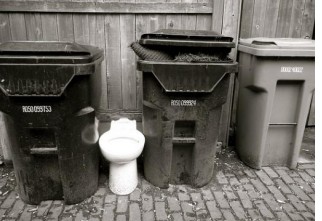 |
| ***a toilet*** *sits next to a shower an sink*
Aes 4.192, Rel 0.361 | ***a toilet*** *with a wooden seat on top of it*
Aes 5.226, Rel 0.371 | ***a toilet*** *in between two trash cans*
Aes 5.551, Rel 0.434 |
| 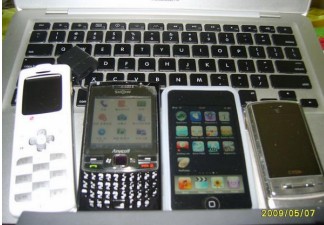 | 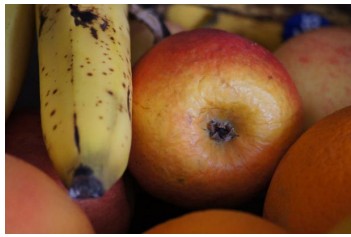 | 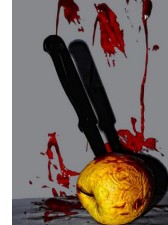 |
| ***an apple*** *phone and some other type of machine*
Aes 3.586, Rel 0.361 | ***an apple*** *and other fruit are sitting together*
Aes 5.007, Rel 0.373 | ***an apple*** *with a knife stuck into it dripping blood*
Aes 5.484, Rel 0.395 |
| 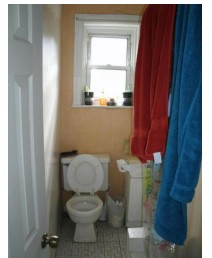 | 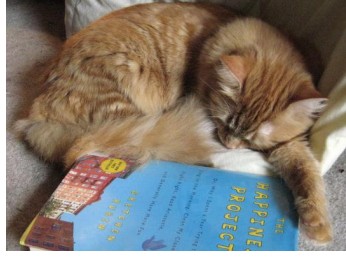 | 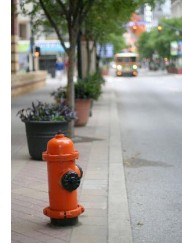 |
| ***an orange*** *and blue bathroom with a tub sink and toilet*
Aes 4.187, Rel 0.357 | ***an orange*** *cat with its eyes closed sitting next to books*
Aes 4.434, Rel 0.359 | ***an orange*** *and black fire hydrant sitting close to a curb*
Aes 5.603, Rel 0.393 |
| 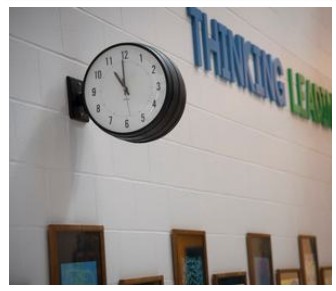 | 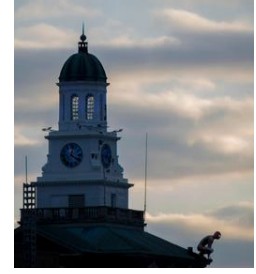 | 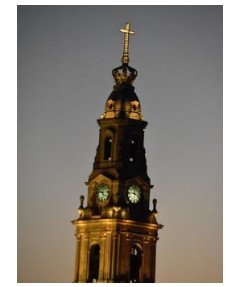 |
| ***a clock*** *on the wall of a room*
Aes 4.578, Rel 0.355 | ***a clock*** *tower with a statue in front of it*
Aes 5.187, Rel 0.376 | ***a clock*** *tower with a cross on top*
Aes 5.425, Rel 0.388 |

