# OpenReview forum: "Seeing Through Words: Controlling Visual Retrieval Quality with Language Models"
_ICLR.cc/2026/Conference — ICLR 2026 Poster_

### Official Review · Reviewer_PhGu · 2025-10-26

**Soundness:** 2
**Presentation:** 3
**Contribution:** 2
**Rating:** 4
**Confidence:** 3

**Summary:**

This paper introduces the problem of Quality-Controllable Retrieval, which is an extension of the text to image retrieval problem where in the input query can have additional constraints such as high relevancy, low aesthetics, etc and the goal is to retrieve images from a gallery which satisfy the original query and the additional constraints as well. To enable existing VLMs such as CLIP to do this, they use a query augmentation pipeline where they train an LLM to generate modified queries based on certain additional constraints (relevancy, aesthetics in their case). The trained LLM is then used during inference time to change the query based on the additional constraints. Experiments on MS-COCO and a new Flickr2.4M dataset show the method can effectively steer retrieval results. Conditioning on "High" quality levels yields images with significantly better average aesthetic and relevance scores compared to "Low" conditions or baseline methods.

**Strengths:**

1. The paper tackles a practical and significant problem. Users frequently use short, ambiguous queries and most existing T2IR systems just return the top-k images by cosine similarity rather than controlling for additional quality metrics. The formulation of "quality-controllable retrieval" is a strong and useful contribution.
2. Simple and elegant solution: their approach allows to convert any existing VLM into a quality controllable one by using a fine-tuned LLM which can augment the input text query to constrain the search.
3. Good visualizations / interpretable results: The paper contains several qualitative examples which show that by fine-tuning the LLM to augment queries, they are able to get precise augmented queries where the retrievals which match the users preferences.

**Weaknesses:**

1. The same CLIP model and EV_A model are used to generate the training data quality labels and for measuring the average metrics during evaluation as well, this might lead to the fine-tuned LLM learning patterns which work only for a particular gallery/dataset and might overfit the query augmentations to that dataset. It would be great if the authors can either do a small human study or use other stronger SoTA multi-modal models as relative judges of the metrics.
2. Missing per-condition re-rank baselines in table 6: the paper only compares with a retrieve -> sort by aesthetics post filtering pipeline in Table 6. This effectively is only covering the H/H sub-segment. Since the main idea behind the paper is QC^2, the table should also include the (L/M/H X L/M/H) segments like in Table 3/4 and also include stronger retrieval baselines like the following:
a) a weighted joint scorer which ranks the images based on a weighted average between rel and aes distance with the queries gt_rel and gt_aes
b) constrained retrieval: since the rel and aes values are already binned, they can first be used to filter out all images which do not belong in the gt bin and then performing search over the reduced sub-space.
These results should help shed more light into how simple retrieval methods perform against more complex query augmentation methods
3. Dataset dependency: As already mentioned by the authors, for each new dataset, they need to fine-tune the query augmentation LLM again so that the LLM can learn the specifics of rel and aes scores and the corresponding captions. This questions the scalability of the method if it needs to be deployed in a real scenario where web-scale retrieval is performed which has a lot of noise/distribution shift.

4.Mode-collapse/diversity risk: QC^2 relies on an LLM to generate quality-conditioned completions, yet the paper reports no diversity or entropy analysis of those texts. Without such checks, the LLM may collapse to a few stylistic templates which led to low loss with a fixed rel and aes value, which can lead to poor diversity in the images which the users would see. It would be great if the authors can provide some metrics on the diversity of queries generated per (rel,aes) tuple. One such metric can be follows:
for a fixed instruction (query: q, rel: r, aes: a), sample n different completions from the llm with different temperature parameters, and then check how diverse these completions are (dispersion based on text-clip distance can be a good metric).

**Questions:**

refer to weaknesses section for questions

---

> ### Author Response · Authors · 2025-11-25
> **Response to Reviewer PhGu (Part 1)**
>
> **Dear Reviewer PhGu,**
>
> ***Thank you very much for recognizing our contributions and for providing constructive comments.*** We address your concerns as below.
>
> **W1 Other multi-modal models as judges.**
>
> Thank you for this constructive comment. Following your suggestions, we conducted additional experiments using independent evaluation models that were never used during training.
>
> We first revisited the MSCOCO dataset and evaluated aesthetic quality using an additional aesthetic evaluation model, LAION-Aesthetics Predictor (sa_0_4). Note that this model is only used for testing. The comparison results are shown below:
>
> ||low|median|high|
> |--|--|--|--|
> |model used in paper|4.74±0.371|4.91±0.440|5.115±0.439|
> |new model|5.352±0.952|5.554±0.963|5.754±0.879|
>
> For further validation, we conduct a second study on the VisualNews dataset, which contains 1M image-text pairs. We consider image quality assessment (IQA) models for quality evaluation, as there are many open-sourced models that can be used for evaluation. During training, we use a multimodal LLM, DeQA-Score [a1], to calculate the quality scores and design the following instruction to train our completion model:
>
> “DeQA Quality: l(score_i), Query: ”.
>
> After training, we evaluate retrieval quality using two additional IQA models: MANIQA [a2], and CLIP-IQA [a3], which again were not involved in training. The experimental results are shown below:
>
> ||low|median|high|
> |--|--|--|--|
> |Avg DeQA-Score|3.348|3.549|3.621|
> |Avg MANIQA|0.497|0.526|0.534|
> |Avg CLIP-IQA|0.630|0.648|0.657|
>
> In all cases, the retrieved images exhibit progressively higher quality as the requested level increases.
>
> **W2 Re-rank baseline**
>
> Thank you for this valuable suggestion. We follow your recommendation and include all nine (L/M/H X L/M/H) quality segments.
> For the re-rank baseline, we first retrieve images based on their top-K minimal/median/maximum relevance scores and then re-order these candidates according to minimal/median/maximum aesthetics scores. This yields results for all nine quality combinations.
>
> For the constrained retrieval baseline, we partitioned the gallery into the nine (L/M/H×L/M/H) segments using their pre-assigned relevance and aesthetic bins, and performed retrieval only within the corresponding segment for each target quality combination.
>
> Regarding the proposed weighted joint scorer, we would appreciate a bit more clarification. In text-to-image retrieval, the basic ranking is determined by the similarity between the textual query and image embeddings, which reflects semantic relevance. It is unclear how to retrieve images with a weighted score. If the reviewer could provide additional details about the intended formulation, we would be happy to include this variant in our experiments.
>
> We evaluate all baselines with K=10 on the MSCOCO dataset. The comparison results are shown below.
>
> ||Aes Cond|L|L|L|M|M|M|H|H|H|
> |--|--|--|--|--|--|--|--|--|--|--|
> ||Rel Cond|L|M|H|L|M|H|L|M|H|
> |rerank|Aes|	4.709|4.330|4.219|5.256|4.931|4.804|5.862|5.626|5.394|
> ||Rel|0.004|	 0.113|0.339|0.004|0.113|0.336|0.005|0.113|0.337|
> |constrained|Aes| 	4.481|4.492|4.489|4.974|4.972|4.973|5.375|5.387|5.382|
> ||Rel|0.327| 	0.328|0.327|0.3259|0.326|0.326|0.319|0.320|0.320|
> |Ours|Aes| 	4.232|4.327|4.195|4.853|4.911|4.845|5.518|5.520|5.562|
> ||Rel|0.334|	0.344|0.356|0.338|0.348|0.356|0.337|0.349|0.360|
>
> As shown, while re-rank and constrained retrieval can adjust aesthetic levels through sorting and filtering, they consistently fail to retrieve images that maintain high semantic relevance to the input queries.
>
> **W3 Dataset dependency**
>
> We appreciate the reviewer’s concern and would like to clarify an important point: **retraining is not always required by our method**. Whether a model must be fine-tuned for a new gallery depends entirely on the nature of the quality metric being controlled, not on the retrieval gallery itself.
>
> **1. Dataset-dependent quality metrics (e.g., relevance).**
>
> Some quality metrics, such as relevance, are intrinsically tied to the underlying dataset because relevance scores change when the image gallery changes. In these cases, fine-tuning is indeed necessary. Any method that aims to predict or control relevance must be aligned with the specific gallery. **This is a property of the metric, not a limitation of our framework.**
> Moreover, even if retraining is required, it is performed entirely *offline*. Once training is completed, the model supports efficient online inference, as demonstrated in our experiments reported in response to Reviewer 7fCB (W2: Inference Latency).
>
>
> *Please refer to the next page for the continuation of our responses.*

---

> > ### Author Response · Authors · 2025-11-25
> > **Response to Reviewer PhGu (Part 2)**
> >
> > **2. Dataset-agnostic metrics (e.g., aesthetics, cleanliness).**
> >
> > Other quality dimensions generalize across datasets and do not depend on the particular image gallery. For such metrics, **a single trained model can be reused without retraining**, even if the retrieval database changes.
> > In Table 4, we show that a model fine-tuned on Flickr2.4M (FT-Blip2) successfully controls aesthetics on MS-COCO without any additional training. This demonstrates that the approach can operate in a plug-and-play manner when the target metric is dataset-agnostic.
> >
> > In short, **retraining is only necessary when the quality metric itself depends on the dataset**. For many commonly dataset-agnostic quality dimensions, the same fine-tuned model can be applied to any gallery.
> >
> > **W4 Collapse and diversity**
> >
> > Thank you for this insightful and constructive comment. We follow your suggestions and conduct additional experiments on the diversity.
> >
> > For a fixed base query (q=’a person’), we test three conditions: (query: q, rel: low, aes: low),  (query: q, rel: median, aes: median), and  (query: q, rel: high, aes: high) for simplicity. For each condition, we sample 100 completions from the LLM using temperature parameters generated by temperatures = np.linspace(0.1, 2.0, 100). We then computed the dispersion of the resulting completions with retrieved images in CLIP embedding space. We compare the minimum, maximum, mean, and std of the CLIP scores, and compare with a pretrained GPT-2 model. The results are shown below:
> >
> > |model|condition|min|max|mean|std|
> > |--|--|--|--|--|--|
> > |pretrained GPT2|high| 0.229| 0.359| 0.298| 0.022|
> > |ours|  high|                   0.347| 0.478| 0.393| 0.019|
> > |pretrained GPT2| median|  0.220| 0.370|  0.286|  0.029|
> > |ours|  median| 0.321|  0.419|   0.369|  0.018|
> > |pretrained GPT2| low|  0.246| 0.354| 0.297| 0.022|
> > |ours|  low|  0.306| 0.414| 0.357| 0.016|
> >
> > Our results show that although our model’s dispersion is slightly lower than that of the pretrained baseline, it nonetheless maintains a meaningful degree of diversity across sampled completions.
> >
> > *We hope the above response adequately addresses your concerns. Once again, thank you so much for your thoughtful feedback and for helping us improve the quality of our paper.*
> >
> >
> > **Appendix:**
> >
> > [a1] Zhiyuan You, et al, Teaching Large Language Models to Regress Accurate Image Quality Scores Using Score Distribution, CVPR 2025.
> >
> > [a2] Sidi Yang, et al, MANIQA: Multi-dimension Attention Network for No-Reference Image Quality Assessment, CVPR 2022.
> >
> > [a3] Jianyi Wang, et al, Exploring CLIP for Assessing the Look and Feel of Images, AAAI 2023.

---

### Official Review · Reviewer_syQK · 2025-10-29

**Soundness:** 2
**Presentation:** 3
**Contribution:** 2
**Rating:** 4
**Confidence:** 4

**Summary:**

This paper proposes Quality-Controllable Retrieval (QCR), which allows users to control retrieval results through quality dimensions such as relevance and aesthetics. The proposed approach uses an LLM as a query completion function conditioned on discretized quality levels. Experiments on multiple datasets demonstrate that quality-aware query completions can enhance retrieval performance, although only two quality metrics are explored.

**Strengths:**

- The paper has a good motivation, addressing the challenge of underspecified queries that often lead to ambiguous retrieval results. Quality-controllable retrieval is both practically useful and of research interest.
- The paper is well-structured, and the methodology is described in sufficient detail.

**Weaknesses:**

- While the concept of quality-controllable retrieval is attractive, the current work only explores two quality dimensions (relevance and aesthetics). This limited scope is insufficient to demonstrate the true practical value of controllable retrieval in applications.
- Despite the claim that LLM-based completions avoid irrelevant or hallucinated content, there is no explicit mechanism or evaluation to manage or detect query artifacts that could mislead retrieval or introduce out-of-distribution details.
- The use of large language models for query rewriting or expansion is already a common and mainstream approach. The main innovation of this paper lies in introducing two quality indicators and training data to guide LLM-based query completions. My primary concern is that this level of novelty does not meet the expected standards of ICLR.

**Questions:**

- The training data consists of concise sentence summarizing the main content of the image with annotated aesthetic and relevance levels. Through training, the LLM learns the relation between text descriptions and the two quality metrics. However, I am concerned **whether such concise sentences are sufficient for the LLM to truly capture the connection between these quality metrics and the actual visual content of the image.**

- Although the numerical results suggest separability, they do not always match intuitive perception. For example, in Table 1 (teddy bear), the left image (aesthetic 4.788, relevance 0.359) seems more aesthetically pleasing and more representative of a teddy bear than the right image (aesthetic 5.818, relevance 0.437). This raises concerns about the reliability of the scoring process.

- The authors claim that the method can be extended to many different metrics. However, the paper only provides experiments with relevance and aesthetics.  **I wonder how the LLM behaves when more metrics (three or more) are involved. Would the model prioritize certain metrics while ignoring others? Will the LLM's completion ability degrade under multiple metrics**？
- Using LLM for query completion is a clever choice, but while the authors claim to avoid hallucinations, no explicit mechanism is provided. Have the authors  observed redundant, voague, or inaccurate queries, and were any filtering strategies considered?

---

> ### Author Response · Authors · 2025-11-25
> **Response to Reviewer syQK (Part 1)**
>
> **Dear Reviewer syQK,**
>
> ***Thank you very much for recognizing our contributions and for providing constructive comments.*** We address your concerns as below.
>
> **W1 and Q3: More quality dimensions.**
>
> Thank you for the constructive suggestion.  As mentioned in Sec. 3.1, the notion of “quality” is flexible and can be instantiated with any metric that is meaningful for a given application domain. We use relevance and aesthetics as examples to show the effectiveness of our method.
>
> To further address the reviewer’s concern on more quality dimensions, we consider an additional quality metric by using the DeQA-Score [a1] model, which is a multimodal LLM for image quality assessment (IQA). Using the MS-COCO dataset, we calculate DeQA-Scores for the gallery images, and train our model with instructions that include three quality dimensions: relevance, aesthetics, DeQA Quality. We test three different instructions, each presenting the metrics in a different order.
>
> 1. “Relevance: l(s^r_i), Aesthetic: l(s^a_i), DeQA Quality:  l(s^d_i), Query:”
>
> |iqa|aes|rel|iqa|aes|rel|iqa|aes|rel|
> |--|--|--|--|--|--|--|--|--|
> |low|low|low|low|low|median|low|low|high|
> |3.639|4.886|0.360|3.677|4.863|0.363|3.752|4.848|0.386|
> |low|median|low|low|median|median|low|median|high|
> |3.733|4.935|0.360|3.670|4.803|0.369|3.741|4.907|0.383|
> |low|high|low|low|high|median|low|high|high|
> |3.734|4.989|0.358|3.733|4.985|0.366|3.549|5.084|0.396|
> |median|low|low|median|low|median|median|low|high|
> |3.821|4.833|0.36|	3.743|4.822|0.365|3.727|4.778|0.381|
> |median|median|low|median|median|median|median|median|high|
> |3.723|4.842|0.357|3.807|4.906|0.366|3.823|4.862|0.382|
> |median|high|low|median|high|median|median|high|high|
> |3.761|4.987|0.360|3.798|4.978|0.371|3.858|4.982|0.384|
> |high|low|low|high|low|median|high|low|high|
> |3.854|4.783|0.357|3.930|4.879|0.367|3.891|4.838|0.386|
> |high|median|low|high|median|median|high|median|high|
> |3.913|4.861|0.362|3.842|4.834|0.365|3.788|4.855|0.383|
> |high|high|low|high|high|median|high|high|high|
> |3.876|5.051|0.354|3.944|5.004|0.364|4.019|5.143|0.393|
>
> Average results:
>
> ||low|median|high|
> |--|--|--|--|
> |avg iqa|3.692|3.785|3.895|
> |avg aes|4.837|4.867|5.023|
> |avg rel|	0.359|0.366|0.386|
>
> 2. “Aesthetic: l(s^a_i), DeQA Quality:  l(s^d_i), Relevance: l(s^r_i), Query:”
>
> |iqa|aes|rel|iqa|aes|rel|iqa|aes|rel|
> |--|--|--|--|--|--|--|--|--|
> |low|low|low|low|low|median|low|low|high|
> |3.764|	4.846|	0.355|	3.693|	4.876|	0.375|	3.672|	4.763|	0.392|
> |low|median|low|low|median|median|low|median|high|
> |3.657|	4.862|	0.352|	3.683|	4.943|	0.374|	3.635|	4.859|	0.392|
> |low|high|low|low|high|median|low|high|high|
> |3.726|	4.995|	0.353|	3.840|	5.039|	0.374|	3.673|	5.043|	0.391|
> |median|low|low|median|low|median|median|low|high|
> |3.837|	4.877|	0.358|	3.838|	4.725|	0.370|	3.791|	4.697|	0.399|
> |median|median|low|median|median|median|median|median|high|
> |3.755|	4.962|	0.351|	3.800|	4.933|	0.373|	3.937|	4.944|	0.396|
> |median|high|low|median|high|median|median|high|high|
> |3.785|	5.017|	0.353|	3.861|	5.065|	0.371|	3.901|	5.103|	0.396|
> |high|low|low|high|low|median|high|low|high|
> |4.028|	4.941|	0.352|	3.858|	4.819|	0.372|	3.977|	4.718|	0.401|
> |high|median|low|high|median|median|high|median|high|
> |3.862|	4.892|	0.352|	3.858|	4.865|	0.371|	3.983|	4.909|	0.395|
> |high|high|low|high|high|median|high|high|high|
> |3.865|	5.077|	0.353|	4.009|	5.128|	0.373|	4.019|	5.138|	0.393|
>
> Average results:
>
> ||low|median|high|
> |--|--|--|--|
> |avg iqa|	3.705|	3.834|	3.940|
> |avg aes|4.807|	4.908|	5.067|
> |avg rel|	0.353|	0.373|	0.395|
>
>
> 3. “DeQA Quality:  l(s^d_i), Relevance: l(s^r_i), Aesthetic: l(s^a_i), Query:”
>
> |iqa|aes|rel|iqa|aes|rel|iqa|aes|rel|
> |--|--|--|--|--|--|--|--|--|
> |low|low|low|low|low|median|low|low|high|
> |3.756|	4.791|	0.358|	3.753|	4.753|	0.366|	3.787|	4.848|	0.382|
> |low|median|low|low|median|median|low|median|high|
> |3.808|	4.975|	0.357|	3.757|	4.918|	0.369|	3.736|	4.920|	0.384|
> |low|high|low|low|high|median|low|high|high|
> |3.700|	5.027|	0.359|	3.766|	4.989|	0.367|	3.603|	5.114|	0.385|
> |median|low|low|median|low|median|median|low|high|
> |3.810|	4.826|	0.360|	3.850|	4.885|	0.370|	3.855|	4.804|	0.383|
> |median|median|low|median|median|median|median|median|high|
> |3.779|	4.912|	0.355|	3.854|	4.891|	0.373|	3.977|	4.915|	0.378|
> |median|high|low|median|high|median|median|high|high|
> |3.891|	5.051|	0.355|	3.848|	4.940|	0.371|	3.843|	5.094|	0.379|
> |high|low|low|high|low|median|high|low|high|
> |3.846|	4.839|	0.356|	3.973|	4.927|	0.371|	3.890|	4.828|	0.386|
> |high|median|low|high|median|median|high|median|high|
> |3.895|	4.874|	0.354|	3.923|	4.937|	0.369|	3.884|	4.860|	0.381|
> |high|high|low|high|high|median|high|high|high|
> |3.856|	4.950|	0.357|	3.953|	5.088|	0.370|	3.926|	5.096|	0.386|
>
> Average results:
>
> ||low|median|high|
> |--|--|--|--|
> |avg iqa|3.741|3.856|3.905|
> |avg aes|4.833|4.911|5.039|
> |avg sim|	0.357|0.370|0.383|
>
> *Please refer to the next page for the continuation of our responses.*

---

> > ### Author Response · Authors · 2025-11-25
> > **Response to Reviewer syQK (Part 2)**
> >
> > As seen from the tables, across all settings, the model consistently achieved controllable retrieval along all three quality dimensions. We did not observe any tendency for our model to favor a particular metric or ignore others, nor did we find evidence that incorporating a third metric compromises the model’s completion ability. These results suggest that our framework behaves robustly when multiple control signals are provided.
> >
> > **W2 & Q4: Inaccurate queries.**
> >
> > Thank you for the insightful comment. In the Supplementary Sec. A.6., we report representative **failure cases in Table 14**, where inaccurate or overly specific completions may influence retrieval. However, such cases are relatively rare in our experiments.
> >
> > More importantly, we would like to clarify that minor textual artifacts are tolerable as the retrieval system is robust. To mitigate artifact issues, we explored simple yet effective strategies: **sampling multiple candidate completions with varying temperature parameters** and allowing the user to choose the preferred one. This not only reduces the risk of relying on a single erroneous completion but also gives users control when their intent cannot be easily expressed through a single query. In many situations, especially when users lack detailed knowledge about the dataset or are unsure how to articulate their preferences, these query suggestions can actually be beneficial.
> >
> > **W3: Novelty.**
> >
> > We appreciate the opportunity to clarify the core novelty of our work, as our contributions extend beyond the use of LLMs for query rewriting,
> >
> > 1. A new problem. The central innovation of our paper lies in formulating and addressing **a new retrieval problem: quality-controllable retrieval (QCR)**. To the best of our knowledge, this problem has not been explored in prior literature. Existing retrieval frameworks focus almost exclusively on improving semantic relevance, whereas our work shows that retrieval can be explicitly guided by additional quality conditions such as aesthetics. Importantly, we demonstrate that this problem is **solvable** and such control can be achieved through a simple method.
> >
> >
> > 2. A new method. Our method solves this new problem by **transforming visual-grounded quality information into language-level instructions** and training an LLM incorporating these quality conditions during query completion. While LLM-based rewriting itself is common, our use of LLMs as a mechanism for **quality conditioning is fundamentally different** from simple expansion. This quality-aware conditioning process is, to our knowledge, unexplored in prior retrieval systems.
> >
> >
> > 3. Theoretical and empirical contributions. We provide both **theoretical analysis and extensive empirical evidence** demonstrating how quality-controllable retrieval is achievable and how well our method works. The theoretical perspective offers insight into how completion can induce more diverse and independent scoring patterns, and our experiments substantiate the practical effectiveness of the proposed method across multiple datasets and quality metrics.
> >
> > We hope that this clarification makes the novelty and significance of our contributions more explicit. ***We believe that defining a new retrieval paradigm, showing its feasibility, and providing an effective solution aligns well with the level of innovation expected for an ICLR contribution***.
> >
> > **Q1: Connection between quality and visual content.**
> >
> > Good question. We’d like to clarify a possible misunderstanding: the **connection** between quality metrics and the actual visual content of the images **is established by the quality evaluation model** (e.g., aesthetics assessor), not by our query completion model.  That means, the quality evaluation model determines how faithfully the quality metrics capture the image’s visual content.
> >
> >
> > The role of our LLM is not to infer visual quality directly from pixels, but to learn how different textual expressions correlate with the quality annotations produced by these external models. During training, the LLM only observes image captions paired with their quality labels. This allows the model to learn linguistic patterns that correspond to lower- or higher-quality images as judged by the used quality evaluators. Because the quality metrics themselves are grounded in visual content through their respective assessment models, the LLM does not need to “see” the images.
> >
> > We emphasize that **as long as these models provide reliable quality annotations, the captions are sufficient for the LLM to learn how to condition query completions on the desired quality levels.**
> >
> > *Please refer to the next page for the continuation of our responses.*

---

> > > ### Author Response · Authors · 2025-11-25
> > > **Response to Reviewer syQK (Part 3)**
> > >
> > > **Q2: reliability of scoring model.**
> > >
> > > This is a good observation. We agree that, when focusing solely on the teddy bear itself, the left image may appear more aesthetically pleasing to human viewers. However, **the aesthetic scores in our experiments are determined by the aesthetic evaluation model, which assesses the entire image composition from overall scene rather than the quality of a single object**. The model we use assigns higher scores to visually striking or less conventional scenes. In this example, the bears in the second and third images appear in more unusual or attention-grabbing environments (e.g., on a lush green tree or beside a stone cabin in a grassy field), which the evaluator interprets as more aesthetically distinctive. By contrast, the first image presents a familiar indoor setting that, although intuitive and natural, is considered less visually remarkable by the aesthetic model.
> > >
> > > If we adopt a quality evaluator specifically tailored to object-level aesthetics, the scoring would likely align more closely with the reviewer’s intuition.
> > > We would like to clarify that **our goal in this work is not to propose a quality evaluator**, but rather to demonstrate that once a quality metric is defined (whether object-centric, scene-centric, or based on any other criterion), our framework can effectively incorporate it into controllable retrieval. The choice of scoring model is also not the focus of our work. Our method is designed to be flexible: **if a stronger quality evaluator is available, our controllable retrieval performance would improve accordingly.**
> > >
> > > *We hope the above response adequately addresses your concerns. Once again, thank you so much for your thoughtful feedback and for helping us improve the quality of our paper.*

---

### Official Review · Reviewer_e8i9 · 2025-10-31

**Soundness:** 3
**Presentation:** 3
**Contribution:** 2
**Rating:** 6
**Confidence:** 4

**Summary:**

This paper proposes a new method for text-to-image retrieval called quality-conditioned query completion . The method addresses the problem of short and underspecified user queries. It uses a large language model to expand short queries with descriptive and quality-aware details. The approach does not modify pretrained vision–language models and provides interpretable, controllable retrieval behavior. Experiments on MS-COCO and Flickr2.4M show that QC² consistently improves both semantic relevance and visual quality, outperforming existing baselines.

**Strengths:**

1. The paper proposes a new task: quality-controllable text-to-image retrieval, which considers both semantic relevance and aesthetic quality during retrieval. This setting aligns well with real-world search scenarios.
2. The method uses a large language model to generate query expansions with quality cues, requiring no modification to existing vision–language models and remaining relatively simple to implement.

**Weaknesses:**

1. The paper mainly focuses on short queries but does not evaluate on datasets with long or descriptive queries. When the textual input already contains sufficient information, the effect of query completion may diminish or even introduce redundancy or semantic drift. Will this strategy still work for underspecified long user query?
2. The paper's quantitative evaluation relies on a limited set of test queries, namely 80 concrete object nouns. Consequently, the method's performance in handling more abstract, complex, or queries involving emotional atmospheres (such as "a sense of calm"，"a vintage vibe") remains entirely unvalidated.
3. This method requires fine-tuning a large language model for each specific image retrieval gallery. This means it is not a "plug-and-play" solution. On the contrary, if the gallery is replaced or updated, the dataset must be rebuilt and significant resources must be invested in retraining, which severely limits its scalability.
4. How can this be extended to image-to-text retrieval?

**Questions:**

1. How does the method perform on the datasets from the LoTLIP[1]?
2. How does the scalability of the QCR method which requires an LLM to be fine-tuned for each gallery, comparing to VISA [2], a "plug-and-play" method that performs re-ranking at test-time without training?

[1] LoTLIP: Improving Language-Image Pre-training for Long Text Understanding

[2] Visual Abstraction: A Plug-and-Play Approach for Text-Visual Retrieval

---

> ### Author Response · Authors · 2025-11-25
> **Response to Reviewer e8i9 (Part 1)**
>
> **Dear Reviewer e8i9,**
>
> ***Thank you very much for recognizing our contributions and for providing constructive comments.***
> We address your concerns as below.
>
> **W1: Longer queries**
>
> For a detailed discussion on long queries, we respectfully refer the reviewer to our response to **Reviewer 7fCB (W1 & Q1: Longer queries with multiple objects and relations)**.
>
> **W2: Abstract queries involving emotional atmospheres**
>
> Thank you for this insightful suggestion. Following this comment, we conducted additional experiments specifically targeting **abstract and emotion-driven queries**. We used GPT5.1 to generate 100 negative and 100 positive abstract queries involving emotional or atmospheric descriptions, which are listed at the end of our response. Using these queries, we evaluated our model on MSCOCO under three different quality levels: low, median, and high. The results are shown below:
>
> *Negative queries:*
>
> |aes|rel|aes|rel|aes|rel|
> |--|--|--|--|--|--|
> |low|low|low|median|low|high|
> |4.913|0.349|4.893|0.355|4.811|0.362|
> |median|low|median|median|median|high|
> |4.927|0.348|4.957|0.353|4.915|0.368|
> |high|low|high|median|high|high|
> |5.042|0.348|5.056|0.357|4.918|0.365|
>
> *Positive queries:*
>
> |aes|rel|aes|rel|aes|rel|
> |--|--|--|--|--|--|
> |low|low|low|median|low|high|
> |5.061|0.348|4.978|0.355|4.934|0.365|
> |median|low|median|median|median|high|
> |5.004|0.348|5.008|0.35|4.925|0.366|
> |high|low|high|median|high|high|
> |4.945|0.351|5.058|0.356|4.981|0.364|
>
> As shown, the model appears to have relatively weak control over these queries. We find that many emotional queries (e.g., “a dreamy and surreal atmosphere”, “a vintage and nostalgic vibe”) **inherently contain aesthetic information**. Using such queries may therefore bias results toward aesthetic dimensions.
>
> To avoid this problem,  we performed another round of testing on the VisualNews dataset, and evaluated retrieval quality using **DeQA-Score** [a3], which is a multimodal large language model for image quality assessment. The experimental results are shown below:
>
> Positive:
> |condition|low|median|high|
> |--|--|--|--|
> |Avg DeQA-Score|3.186 ± 0.658|3.227 ± 0.643|3.311 ± 0.694|
>
> Negative:
> |condition|low|median|high|
> |--|--|--|--|
> |Avg DeQA-Score|3.188 ± 0.604|3.273 ± 0.594|3.342 ± 0.606|
>
> These results show clear **quality-aligned control** across conditions, confirming that our method generalizes beyond concrete-object queries and **also handles abstract emotional queries**.
>
> **W3: Retraining and scalability**
>
> We appreciate the reviewer’s concern and would like to clarify an important point: ***retraining is not always required by our method***. Whether a model must be fine-tuned for a new gallery depends entirely on the nature of the quality metric being controlled, not on the retrieval gallery itself.
>
> **1. Dataset-dependent quality metrics (e.g., relevance).**
>
> Some quality metrics, such as relevance, are intrinsically tied to the underlying dataset because relevance scores change when the image gallery changes. In these cases, fine-tuning is indeed necessary. Any method that aims to predict or control relevance must be aligned with the specific gallery. **This is a property of the metric, not a limitation of our framework.**
>
> **2. Dataset-agnostic metrics (e.g., aesthetics, cleanliness).**
>
> Other quality dimensions generalize across datasets and do not depend on the particular image gallery. For such metrics, **a single trained model can be reused without retraining**, even if the retrieval database changes. For example, in Table 4, we show that a model fine-tuned on Flickr2.4M (FT-Blip2) successfully controls aesthetics on MS-COCO without any additional training. This demonstrates that the approach can operate in a plug-and-play manner when the target metric is dataset-agnostic.
>
> In short, **retraining is only necessary when the quality metric itself depends on the dataset**. For many commonly dataset-agnostic quality dimensions, the same fine-tuned model can be applied to any gallery.
>
> **W4: image-to-text retrieval**
>
> Our current work focuses on the text-to-image retrieval setting, consistent with [a2]. Extending the proposed quality-controlled framework to the image-to-text direction is certainly possible, but it requires a different formulation. A natural extension is to fine-tune a VLM on our quality-augmented image–caption pairs, where the quality information (e.g., aesthetics, clarity, style) is embedded into the textual side or the VLM’s joint representation space. However, this setting lies outside the scope of this paper. We view this as a promising direction and plan to explore it in our future work.
>
> *Please refer to the next page for the continuation of our responses.*

---

> ### Author Response · Authors · 2025-11-25
> **Response to Reviewer e8i9 (Part 2)**
>
> **Q1:LoTLIP dataset**
>
> Due to the large scale of the full LoTLIP collection (100M samples), downloading and processing the entire dataset was not feasible within the rebuttal period. However, to provide a meaningful validation, we conducted experiments on a substantial subset of LoTLIP. Specifically, we downloaded 555,292 image–text pairs from the Long Caption–CC (*cc3m_3long_3short_1raw_captions_url.csv*). For this dataset, we used the Qwen2.5 model and the ShortLLA captions as our base text inputs. The results are reported below:
>
> |aes|rel|aes|rel|aes|rel|
> |--|--|--|--|--|--|
> |low|low|low|median|low|high|
> |4.728|0.342|4.813|0.354|4.703|0.374|
> |median|low|median|median|median|high|
> |4.856|0.342|4.834|0.352|4.899|0.376|
> |high|low|high|median|high|high|
> |5.003|0.345|4.978|0.354|5.109|0.377|
>
> Across all conditions, the retrieved images follow the expected quality ordering. This confirms that our method remains effective on the LoTLIP dataset.
>
> **Q2:Scalability comparison with VISA**
>
> Thank you for highlighting this important work. VISA [2] and our method share the high-level intuition of transforming visual information into textual descriptions, but the two methods **target different problem settings and different retrieval goals**. VISA is designed to improve semantic retrieval using off-the-shelf multimodal models at test time, while our method focuses on quality-controlled retrieval to steer results along user-specified quality dimensions.
>
> The two approaches make different trade-offs in training and inference. VISA is entirely training-free while our method requires retraining when the target quality metric is dataset-dependent. But in the inference stage, our method is considerably more lightweight, requiring only a single 0.5B Qwen2.5 model to complete queries. However, VISA relies on a LLaVA-v1.6-34B for description generation, a Qwen2-VL-32B for question generation, and a Qwen2-VL-7B for answer generation. This leads to higher computational overhead, especially for large-scale retrieval scenarios.
>
> In short, **these two methods ultimately target different capabilities, and are therefore complementary to each other**. We will clarify this comparison in the revised manuscript.
>
> *We hope the above response adequately addresses your concerns. Once again, thank you so much for your thoughtful feedback and for helping us improve the quality of our paper.*
>
> **Appendix:**
>
> [a1] LoTLIP: Improving Language-Image Pre-training for Long Text Understanding
>
> [a2] Visual Abstraction: A Plug-and-Play Approach for Text-Visual Retrieval
>
> [a3] Zhiyuan You, et al, Teaching Large Language Models to Regress Accurate Image Quality Scores Using Score Distribution, CVPR 2025.
>
> Here, we present a few examples of negative and positive abstract queries. We will include the full set of abstract queries in the revised manuscript.
>
> **Negative abstract queries:**
> [*a feeling of unease, a sense of impending doom, an atmosphere of quiet dread, a cold and lonely emptiness, a moment filled with anxiety, a heavy sense of isolation, an unsettling quietness,  a bleak emotional landscape,  a feeling of being watched,  a distant and fading hope*]
>
>
> **Positive abstract queries:**
>
> [*a calm and quiet moment,  a soft sense of melancholy,  a dreamy and surreal atmosphere,  a vintage and nostalgic vibe,  a gentle feeling of warmth,  a subtle emotional tension, a peaceful sense of isolation,  an eerie but quiet calm,  a bittersweet memory,  a hazy dreamlike glow*]

---

### Official Review · Reviewer_7fCB · 2025-10-31

**Soundness:** 3
**Presentation:** 3
**Contribution:** 2
**Rating:** 6
**Confidence:** 3

**Summary:**

This paper explores using an LLM to rewrite short text queries into more detailed, quality-aware descriptions, aiming to control the aesthetic and relevance level of retrieved images without modifying the vision-language model. The method learns to expand queries based on quality labels and shows that higher-quality prompts lead to higher-quality retrieval results.

**Strengths:**

1. The motivation is clear. The paper introduces aesthetic cues to explicitly control and improve retrieval quality.
2. The method is plug-and-play and easy to apply in existing systems, requiring no modification to the visual model.

**Weaknesses:**

1. Experiments mainly use single-word, single-object queries（e.g., “a dog”）. Real retrieval queries are usually longer, involve multiple objects and relations. Current setup looks more like keyword/entity retrieval.
2. The paper doesn't report preprocessing cost or inference latency, so it's hard to judge the efficiency of method.
3. The approach assumes the database has many visually similar images with different aesthetic qualities (like Flickr30k/COCO). In many datasets this won’t hold (VisualNews,fashion200k).

**Questions:**

1. In real retrieval scenarios, queries often contain multiple objects, relations, and context. How would the method scale to complex, natural multi-entity queries?
2. How would the method perform when the image databse diversity is limited, or when aesthetic cues are less meaningful (news/fashion domain)?
3. The theory suggests increased rank leads to better discrimination. But multiple  multiple LLM rewrites could also increase rank. On theoretical, what is the concrete advantage of your controlled rewriting over simple rewriting ?

---

> ### Author Response · Authors · 2025-11-25
> **Response to Reviewer 7fCB (Part 1)**
>
> **Dear Reviewer 7fCB,**
>
> ***Thank you very much for recognizing our contributions and for providing constructive comments.*** We address your concerns as below.
>
>
> **W1&Q1: Longer queries with multiple objects and relations**
>
> We would like to clarify that our work focuses on the search problem with short queries, as stated in Sec. 2.1 (line 98). This setting is meaningful for quality-controlled retrieval because **short queries create a large candidate subspace** that contains images of diverse quality levels. Under this condition, we are able to retrieve high-quality results by extending the query with quality-relevant descriptions.
>
> In contrast, longer queries describing multiple objects and relations fall **outside our problem scope**. Please note that long queries inherently produce an **extremely small retrieval subspace**. Given a fixed retrieval system (e.g., OpenCLIP), such queries may correspond to only one or two images in the entire dataset that satisfy all semantic constraints, *making further query extension unnecessary and ineffective*.
>
> To verify this, we generated 100 multi-object relational long queries using GPT5.1 and evaluated our method on MS-COCO. These longer queries can be found at the end of our response. The table below summarizes the scores under different quality conditions:
>
> |aes|rel|aes|rel|aes|rel|
> |--|--|--|--|--|--|
> |low|low|low|median|low|high|
> |5.067|0.341|5.04|0.342|5.059|0.345|
> |median|low|median|median|median|high|
> |5.027|0.341|5.048|0.342|5.052|0.346|
> |high|low|high|median|high|high|
> |5.053|0.341|5.013|0.342|5.064|0.349|
>
> As shown, the score differences across conditions are marginal. When inspecting the retrieved images, we found that ***~60% of the queries produce the same top-1 result images, regardless of the target quality condition*** (e.g., “low-aes, low-rel” vs. “high-aes, high-rel” share 59/100 identical top-1 outputs). This happens because:
> 1) Long multi-entity queries produce extremely limited retrieval subspaces, often containing only one or two semantically matching images (for a given retrieval system). Consequently, any extension of the query maps to the same result regardless of quality conditions.
>
> 2) Completion models trained on MSCOCO tend to generate queries with lengths and structures similar to original captions. For long queries that already match COCO caption lengths and structures, there is little room for meaningful quality-conditioned extension.
>
> For these reasons, we emphasize that long, highly descriptive queries inherently carry sufficient semantic information and therefore **do not benefit from our generation-based completion approach**. Addressing quality control for such complex queries likely requires a different formulation beyond query extension, and we plan to explore these directions in future work.
>
> **W2: Inference latency.**
>
> Thank you for this constructive comment. Following the reviewer’s suggestion, we measured the inference latency of our method on the VisualNews dataset (1M+ images). We evaluated the wall-clock time of the GPT2 completion stage on a single NVIDIA L40S GPU. The table below reports the total time for completing 80 queries under different quality conditions:
>
> |condition|low|median|high|avg(80)|avg(1)|
> |--|--|--|--|--|--|
> |wall-clock|66.284|54.104|49.389|56.592|0.707|
>
> As shown, the completion process is highly efficient: the average time per query is **0.7 seconds**, which is negligible compared to the cost of embedding extraction and nearest-neighbor search in large-scale retrieval systems. We will include these results in the revised manuscript.
>
>
> **W3&Q2: VisualNews & fashion200k datasets.**
>
> Thank you for this insightful comment. In our experiments, we use relevance and aesthetics as two examples to show the effectiveness of our method. But we would like to clarify that **our method does not assume or require aesthetics to be the only or even the primary quality dimension**. As mentioned in Sec. 3.1, *the notion of “quality” is flexible and can be instantiated with any metric that is meaningful for a given application domain*. Aesthetics preference is only one example.
>
>
> To examine how our approach behaves under different quality definitions and in domains where visual diversity or aesthetic cues are limited, we conducted additional experiments on VisualNews and Fashion200k.
>
> ***1. Experiments on VisualNews:***
>
> We download the VisualNews dataset from official github (VisualNews-Repository), which contains more than 1M samples. We define image quality using DeQA-Score [a1], a multimodal large language model for image quality assessment.
>
> We first computed DeQA-Score for all images and trained our model using the following instruction (for simplicity, we do not consider relevance here):
>
> “DeQA Quality: l(score_i), Query: ”,
>
> where the score is the DeQA-Score of the i-th image.
>
> *Please refer to the next page for the continuation of our responses.*

---

> > ### Author Response · Authors · 2025-11-25
> > **Response to Reviewer 7fCB (Part 2)**
> >
> > Below are results on VisualNews with the same testing query objectives used in MS-COCO and Flickr2.4M:
> > |condition|low|median|high|
> > |--|--|--|--|
> > |Avg DeQA-Score|3.414 ±0.605|3.549 ± 0.459|3.621 ± 0.449|
> > |Avg Relevance|0.366 ± 0.020|0.364 ± 0.024|0.362 ± 0.022|
> >
> > The retrieved images follow the intended **DeQA quality ordering**, while relevance remains similar because it was not used as a constraint in this experiment. This demonstrates that our method can steer retrieval according to other quality dimensions.
> >
> > *Additional test: abstract emotional queries (as suggested by Reviewer e8i9)*
> >
> > We further created abstract queries involving emotional atmospheres on this dataset (queries can be seen from our response to Reviewer e8i9). Results again show consistent quality control:
> >
> > |condition|low|median|high|
> > |--|--|--|--|
> > |Positive|3.186 ±0.658|3.227 ± 0.643|3.311 ± 0.694|
> > |Negative|3.188 ±0.604|3.273 ± 0.594|3.342 ± 0.606|
> >
> > These results suggest that our method generalizes well to non-aesthetic quality criteria and various queries.
> >
> >
> > ***2. Experiments on Fashion200K:***
> >
> > We download the Fashion200k data from huggingface (Marqo/fashion200k) for experiments. It contains many near-duplicate product images, where aesthetics and even fine-grained relevance signals are extremely small. In this domain, we define “quality” as product popularity. Because real popularity scores are unavailable, we generate synthetic popularity labels using a truncated normal distribution over [0, 5], derived from a Gaussian with a mean of 2.5 and a standard deviation of 1.0.
> >
> > We train the model with the following instruction:
> >
> > "Relevance:  l(rel_i), Popularity:  l(pop_i), Query:"
> >
> > where rel_i and pop_i are relevance and popularity scores of the i-th image.
> >
> > We designed 50 queries representing combinations of 5 clothing categories [dress, shirt, skirt, pants, jacket] and 10 common colors (e.g., red dress, green jacket, etc.). Results are shown below:
> >
> > |pop|rel|pop|rel|pop|rel|
> > |--|--|--|--|--|--|
> > |low|low|low|median|low|high|
> > |2.242|0.34|2.648|	0.343|2.480|0.345|
> > |median|low|median|median|median|high|
> > |2.452|0.339|2.507|0.346|2.563|0.349|
> > |high|low|high|median|high|high|
> > |2.495|0.353|2.475|0.347|2.542|0.36|
> >
> > The model’s control over popularity and relevance is weaker on Fashion200k. This is expected and likely due to:
> > 1) Many images look very similar; even humans struggle to distinguish them. As a result, extending a query with more details does not meaningfully change retrieval.
> > 2) Off-the-shelf OpenCLIP embeddings cannot effectively differentiate near-duplicate product shots.
> > 3) Many images belong to the same product and should share the same popularity, but our synthetic scores assign random values, which limits learnability.
> >
> > We expect performance to improve if (a) stronger domain-specific VLMs (i.e., the retrieval system) are available to use, and (b) real popularity labels or reliable popularity-prediction models become available.
> >
> >
> > **Q3: Controlled rewriting over simple rewriting.**
> >
> > Our theoretical motivation is that query completion increases the rank of the score matrix by inducing more diverse and independent scoring patterns. We agree that any type of rewriting, such as uncontrolled or simple rewriting may also increase the rank to some extent. However, the core theoretical and practical advantage of our method lies in how the rank is increased and what the additional dimensions encode.Simple rewriting introduces variations that potentially improve semantic alignment with the image content. These variations are not directionally meaningful: they may increase diversity in phrasing, but they do not systematically shift the matching function toward any target property. As a result, the additional rank components mostly reflect linguistic or semantic noise rather than controlled discriminative factors. In contrast, **our controlled rewriting explicitly conditions the completion on image quality variables** (e.g., aesthetics, relevance, DeQA-Score). This conditioning **induces quality-aligned perturbations** in the query space. Thus, the new dimensions encode meaningful information corresponding to the desired quality metric rather than incidental linguistic variation. In our experiments, we tested a pretrained model for simple rewriting, and the results are shown in Tables 3 and 4.
> >
> > *We hope the above response adequately addresses your concerns. Once again, thank you so much for your thoughtful feedback and for helping us improve the quality of our paper.*

---

> ### Author Response · Authors · 2025-11-25
> **Response to Reviewer 7fCB (Part 3)**
>
> **Appendix:**
>
> [a1] Zhiyuan You, et al, Teaching Large Language Models to Regress Accurate Image Quality Scores Using Score Distribution, CVPR 2025.
>
> [a2] Fuxiao Liu, et al, Visual News: Benchmark and Challenges in News Image Captioning, EMNLP 2021
>
> [a3] Xintong Han et al, Automatic Spatially-aware Fashion Concept Discovery, ICCV2017
>
>
> Below, we present a few examples of longer queries with multiple objects and relations. Due to space limitations, we will include the full set of these queries in the revised manuscript.
>
> [*A dog chasing a ball on a beach, Two children feeding ducks at a pond, A woman holding an umbrella in the snow, A cat sleeping beside a stack of books, A boy flying a kite near a lighthouse, Cars parked under blooming cherry trees, A couple taking photos in front of a waterfall, A family watching fireworks beside a river, A group of tourists taking pictures near ancient ruins*]

---

> ### Comment · Reviewer_7fCB · 2025-11-27
> **Post-Rebuttal Feedback**
>
> Thank you for your response. My concerns have been effectively addressed, and I will maintain my original positive score of 6.

---

### Author Response · Authors · 2025-11-26
**General Response to All Reviewers**

**Dear Reviewers**,

***Thank you very much for your thoughtful and constructive comments on our submission!***

We are pleased and encouraged to see a consistent recognition of our contributions across all reviewers:

- **Reviewer 7fCB**: *"clear motivation”, “easy to apply"*

- **Reviewer e8i9**: *"a new task aligns well with real-world search scenarios”, “simple to implement"*

- **Reviewer syQK**: *"good motivation", “practically useful and of research interest”, “attractive concept”*

- **Reviewer PhGu**: *"a practical and significant problem", “strong and useful contribution”, “simple and elegant solution”*


To help re-engage with the paper, we briefly summarize our **core contributions** and our **responses to address the remaining concerns**:

---

**Core Contributions**

1. **A new problem: quality-controllable retrieval**

We explore and formalize a new problem: whether image retrieval results can be explicitly controlled by user-predefined quality constraints? We demonstrate that this problem is solvable in practice.



2. **A simple, general solution: quality-conditioned query completion**

We proposed a simple and broadly applicable method that transforms visual-grounded quality information into language instructions and trains a LLM to generate quality-conditioned query extension, as exactly expressed in our title: *Controlling Visual Retrieval Quality with Language*.


3. **Theoretical analysis and empirical validation**

We provide both theoretical insight and extensive empirical evidence to demonstrate the effectiveness of the proposed method.

---

**Responses to Reviewer Concerns**

To address the concerns raised across the reviews, we have made substantial additions and clarifications, such as:

- Adding new experimental results on VisualNews (1M), Fashion200K, LoTLIP datasets (550K);

- Evaluating performance on longer, multi-entity, abstract, and emotion-driven queries;

- Incorporating an additional quality dimension (DeQA-Score) for further validation;

- Measuring inference latency and completion diversity;

- Clarifying that retraining and scalability essentially depend on the definition of quality metrics;

- Using additional independent scoring models as judges;

- Benchmarking against reranking and constrained retrieval baselines;

- Clarifying points that may have caused misunderstanding.

---

We hope that these clarifications and extended experiments sufficiently address your concerns and reinforce the significance of our contributions.


***We would be very grateful if you could kindly take a moment to review our detailed responses.***

We sincerely appreciate your time and dedication in helping improve the quality of our work!

Warm regards,

The Authors

---

> ### Author Response · Authors · 2025-12-03
> **Rebuttal Summary**
>
> This paper received four reviews. After our detailed rebuttal, ***Reviewer 7fCB confirmed that all of their concerns were fully resolved***. Due to policy-related constraints, the remaining **three reviewers were unable to provide timely follow-up feedback or participate in the discussion**. Nevertheless, we believe we have thoroughly addressed their concerns. We summarize our responses below.
>
> ---
> **Reviewer e8i9**
>
> Main concerns:
> - Test longer queries;
> - Test abstract/emotional queries;
> - Scalability of the method;
> - Extension to image-to-text retrieval;
> - Experiments on LoTLIP dataset;
> - Comparison with VISA baselines.
>
>
> **Our responses:**
> - Added experiments using longer GPT-5.1–generated queries.
> - Added experiments on both positive and negative abstract queries.
> - Clarified that retraining is not always required; scalability limitations arise from the definition of the quality metric, not from our framework.
> - Provided a feasible extension of our method to image-to-text retrieval.
> - Added experiments on the LoTLIP dataset.
> - Compared with the VISA baseline, showing our method and VISA involve different training/inference trade-offs and are complementary to each other.
>
> ---
> **Reviewer syQK**
>
> Main concerns:
> - Need more quality dimensions;
> - Inaccurate queries and hallucinations;
> - Novelty of the work;
> - Connection between quality metrics and visual content;
> - Reliability of the scoring model.
>
>
> **Our responses:**
> - Introduced an additional quality metric (DeQA-Score) and evaluated with three different instructions.
> - Reported representative failure cases (Table 14) and added a simple mitigation strategy using multi-temperature sampling.
> - Clarified our novelty: proposing a new retrieval paradigm, establishing its feasibility, presenting an effective solution (QC²), and providing theoretical and empirical validation.
> - Explained that the quality–content relationship is determined by the external evaluation model, not by our LLM-based query completion model.
> - Clarified that our aesthetics scores reflect full-image composition (not object-level), and our framework does not depend on any specific scoring model.
>
>
> ---
> **Reviewer PhGu**
>
> Main concerns:
> - Evaluate with other multimodal judges;
> - Add re-rank and constrained baselines;
> - Dataset dependency and scalability;
> - Diversity of generated completions.
>
>
> **Our responses:**
> - Evaluated with multiple independent quality models not used in training.
> - Added all 9 quality segments, re-rank baselines, and constrained retrieval baselines.
> - Clarified that retraining is needed only for dataset-dependent metrics (e.g., relevance), while dataset-agnostic metrics (e.g., aesthetics) transfer across datasets without retraining.
> - Conducted temperature-based sampling to assess diversity and confirmed no mode collapse.
>
>
>
>
>
> ---
> ***Overall, although three reviewers were unable to provide follow-up feedback due to policy constraints, we believe our responses have fully resolved all raised concerns***. The issues brought up by the reviewers were addressed comprehensively and consistently in our rebuttal, and the additional results further confirm the soundness and robustness of our approach.

---

### Meta-Review · Area_Chair_LpkN · 2026-01-11

**Summary:**

This paper proposes a quality-controllable text-to-image retrieval paradigm that uses a generative LLM to extend underspecified short queries with quality-aware details, compatible with existing VLMs without modification. Extensive experiments demonstrate the approach’s effectiveness in improving retrieval results and enabling quality control.

Reviewers' main concerns are: unsubstantiated bias-resistance claim, disconnect between proposed system and evaluation, lack of technical novelty, insufficient evaluation depth and generalizability, and inadequate literature review.

**Reviewer Concerns:**

Addressed Concerns:
- Introduced an additional DeQA-Score quality dimension to address the concern of limited quality dimensions.
- Supplemented experiments on long, abstract, and emotional queries to resolve the issue of insufficient evaluation on complex queries.
- Clarified scalability by explaining that retraining is not needed
- Conducted diversity analysis of LLM-generated completions to tackle the lack of diversity analysis.

Outstanding Concerns:
- Despite the authors explicitly clarifying novelty (new quality-controllable retrieval paradigm, quality-conditioned query completion, and theoretical/empirical validation), I didn't agree that the paradigm is totally new. The method novelty is also kind of limited.

**Reviewer Scores:**

- Reviewer 7fCB: Confirmed all concerns were fully addressed, maintaining the original score of 6
- Reviewer e8i9: assuming the rebuttal addressed most concerns, the score is expected to remain 6
- Reviewer syQK: the score is expected to maintain 4 due to the novelty issues.
- Reviewer PhGu: if most concerns are addressed, the score is expected to increase from 4 to 6

---

### Decision · Program_Chairs · 2026-01-26

Accept (Poster)